# Evolution characteristics of temporal and spatial pattern of Russian economic differences since the 21st century

**Nan-Chen Chu**[1], **Ping-Yu Zhang**[2,3]*, **Xiang-Li Wu**[1]*

**1** College of Geographical Sciences, Harbin Normal University, Harbin, China, **2** Northeast Institute of Geography and Agroecology, Chinese Academy Sciences, Changchun, China, **3** University of Chinese Academy of Sciences, Beijing, China

* zhangpy@iga.ac.cn (PYZ); jndxwxl@163.com (XLW)

**Data Availability Statement:** All relevant data are within the paper and its Supporting Information files.

**Funding:** This work is supported by the National Natural Science Foundation of China (No.

## Abstract

Under the background of "the Belt and Road" and "the economic corridor of China, Mongolia and Russia" initiatives, it is of great significance to study the temporal and spatial economic pattern in the Russian Federation. Based on the economic development difference index, regional economic grade index, global trend analysis tool and spatial autocorrelation model, this paper analyzes the temporal and spatial pattern evolution characteristics of Russian economic differences from 2002 to 2020. The results are as following. First, although the economic imbalance among various federal subjects has been decreasing, the economic polarization has been still severe between the prosperous developed regions and the stagnant backward regions during 2002–2020. Russia's economy shows a trend of changing from significant positive correlation in strong agglomeration space to positive correlation in weak agglomeration space, and then to random distribution. Second, there has been great differences of the economic development among various federal subjects. The economic grade of the Russian federal subjects presents a significant spatial differentiation pattern. The Russian Federation's economic resources are concentrated in the first-class federal subject (Moscow City), second-class federal subjects (Tumen Region, Moscow Region and Saint-Petersburg city) and a few third-class federal subjects (Yamalo-Nenetsky Autonomous Area, Khanty-Mansiysky Autonomous Area, Republic of Tatarstan, Krasnodar Territory, Sverdlovsk Region, etc). Third, the Russian Federation's economy presents "High Core, Low Periphery", "High West, Low East" and "High south, Low north" spatial differentiation pattern. The economic hot regions coincide with the high-class economic regions, which are mainly distributed in the contiguous areas of Ural Federal District and Volga Federal District, as well as the Moscow City, Moscow Region, Saint-Petersburg city, Krasnodar Territory and Rostov Region. The economic cold regions coincide with the low-class economic regions, which are mainly located in the Far East Federal District, the east of Siberian Federal District, the north of North West Federal District and the south of North-Caucasian Federal District. Finally, we suggest the recommendation for policy makers in Russia. And we propose the future research ideas.

42101165; 42071162), Heilongjiang Philosophy and Social Sciences Research Planning Project (No. 21JLC201), China Postdoctoral Science Foundation (No. 2021M693817), and Heilongjiang Postdoctoral Science Foundation (No. LBH-Z21067).

**Competing interests:** The authors have declared that no competing interests exist.

## Introduction

The regional economic development level is defined as the development trend of regional economic structure, quantity, quality, speed, and other economic aspects [1]. It is closely related to regional economic foundation and regional development stages [2]. Affected by such factors as location, resources, technology, policies, industrial structure, market development degree, and opening up to the outside world, different regions have different economic development levels. Regional economic development difference is an objective phenomenon formed by the interaction and correlation of different economic elements. Appropriate regional economic differences are conducive to the flow of economic elements, the effective allocation of resources and the transfer of industrial space, enabling each region to play its comparative advantages through competition and cooperation [3]. However, excessive regional economic differences could cause regional economic imbalances, a series of political and social problems. And they eventually affect the stability and coordinated development of the whole country [4].

The Russian Federation is the country with the largest land area, the richest mineral and energy resources in the world [5]. After the disintegration of the Soviet Union, the Russian Federation's economic development went through five important stages [6]. (1) The first stage of Russian Federation's economic development was the severe economic and social crisis stage in 1992–1998. After independence from the Soviet Union, the Russian Federation faced one trillion rubles of domestic debt and 100 billion dollars of foreign debt [5]. The "shock therapy" implemented by Yeltsin made a serious of strategic mistakes when the Russian Federation changed from planned economy type to market economy type. Radical reforms caused lots of problems, including a sharp increase in unemployment rate, and a vicious circle of inflation in the Russian Federation. The Russian Federation's "October Event" broke out in 1993. Then the first Chechen War erupted during 1994–1996. From 1997 to 1999, four prime ministers were replaced continuously. These events brought heavy damages to the political and economic development of the Russian Federation. (2)The second stage of Russian Federation's economic development was the rapid economic growth stage in 1998–2007. The Russian Federation's economic crisis ended in August 1998. And its export price of mineral resources increased significantly from March to April in 1999. In August 1999, Putin was appointed as the prime minister of the Russian Federation. The Russian Federation's economy gradually recovered because of the stable political environment, efficient central management, and good international market. (3) The third stage of Russian Federation's economic development was the economic crisis stage caused by the global financial crisis in 2007–2010. Medvedev won the presidential election of the Russian Federation in 2008. The primary task faced by Medvedev and Putin was to comprehensively improve the national economic systems. However, this process was interrupted by the Russian-Georgian war and the global financial crisis. In particular, the physical economy of the Russian Federation suffered heavy losses due to the collapse in energy prices. (4) The fourth stage of Russian Federation's economic development was the economic recovery and micro-growth stage of 2010–2013. The anti-economic crisis measures which were put forward by the Russian federal government took effect in 2010. The Russian Federation's economy recovered to its pre-economic crisis level in 2011. Its economy development maintained slow growth during 2012–2013. Its average annual growth rate of GDP was 2.8%, and the average annual growth rate of investment was 3.6% during 2010–2013 [7]. (5) The fifth stage of Russian Federation's economic development was the domestic economic crisis stage imposed by European and American sanctions from 2014 to the present. Under the influence of falling oil prices, the Ukraine crisis and the multiple rounds of economic sanctions from the Europe and United States, Russian Federation's economic weakness had been more severe. The Russian Federation's GDP increased by 0.6% in 2014 compared with 2013. Its total GDP reduced by 3.7%-3.8%, the inflation rate was higher to 12.9%, the import and export

volume decreased by 33.2% in 2015 compared with 2014 [8]. The economic development gravity of the Russian Federation began to gradually turn to the eastern countries, such as China.

The Russian Federation is an important neighboring country of China. Researches on Russia's economic development focus on the economic development situation, economic development differences, economic macroeconomic pattern, economic development characteristics, economic development trends, industrial development and reindustrialization of specific federal districts and federal subjects. Fedorov combined with Gini coefficient, GE index, ER index, Wolfson index and other indexes to study the polarization trend of Russian economic development from 1990 to 1999 [9]. Based on the hypothesis of spatial equilibrium and agglomeration economy, Kolomak believed that the Russian Federation had high spatial heterogeneity of economic activities, showing the agglomeration development. And the agglomeration speed in the west Europe was stronger than that in the east Asia [10]. Vertakova pointed out that the asymmetry of Russia's economic development was gradually weakening, but the economic imbalance degree was still high in the Russian Federation [11]. Granberg diagnosed the regionalization of the efficiency of Russia's economic anti-crisis planning, and he discussed the possible scenarios and future trends of Russia's economic recovery [12]. Taking the Siberian Federal District and the Far East Federal District as examples, Seliverstov compared competitive potential of labor and investment resources in economic development [13]. And from the efficiency of mineral and raw material development projects, Glazyrina studied the long-term economic benefits of cross-border cooperation between Russia and China [14]. Kuleshov discussed the development direction of reindustrialization planning in Novosibirsk, proposing the most competitive reindustrialization strategic measures with scientific innovation, engineering and manufacturing [15]. Kuz'minov studied the economic problems and social impact of the wood industry system in Kostroma Region, proposing its strategic countermeasures to adapt to the economic recession in 2009 [16]. The main conclusions drawn from the previous literatures were as followed. Since the period of economic transition, the uneven spatial allocation of industrial activities aggravated the polarization of regional economic and social development in the Russian Federation. And the economic differences among various regions had been increasing both qualitatively and quantitatively in the Russian Federation. This phenomenon had seriously restricted Russia's market reform and economic growth. The proportion of primary, secondary and tertiary industries was unbalanced in the Russian Federation, showing that the heavy industry was too heavy, the light industry was too light, the agriculture and the modern service industry fell behind for a long time. In addition, Russia's scientific and technological contribution rate was low, and Russia's economic development still depended on the labor and material capital investment. The economic growth rate of different Russian federal subjects was also unbalanced, which was reflected in the contraction of economic living space. In the future, the development of the Far East Federal District and the North-Caucasian Federal District will play a important role in promoting the regional economic balance in the Russian Federation [9–16].

Under the background of the rapid development of economic globalization, bilateral relations between China and Russia have reached a high level of cooperation. In 1996, China and Russia established a strategic cooperative partnership. In 2001, China and Russia signed the Sino-Russian treaty of good neighborliness, friendship and cooperation. In 2013, China and Russia established a new win-win cooperation relationship under the background of "the Belt and Road" initiative. The trade intensity, import and export volume between China and Russia had steadily increasing. In 2020, the trade volume between China and Russia reached 107.8 billion dollars. China has become Russia's largest trading partner for many years. The Russian Federation has formed a primary product export structure dominated by energy minerals to China [5, 17]. China has also formed a higher product export structure of machinery

manufacturing, textile clothing and metal products to the Russian Federation [17, 18]. At the same time, the traditional commodity services have expanded to the science-technology, transportation, tourism, military, environmental protection, energy and other fields in the Xinjiang-Western Siberia Federation district, Northeast China-Far East Federation district, and Northeast China-Siberian Federation district [18–20]. But unfortunately, the Corona Virus Disease 2019 (COVID-19) affected the overall economic activities between China and Russia. What should China and Russia do in the epidemic prevention and economic trade cooperation? The new topics are put forward for the scholars. Therefore, studying the temporal and spatial pattern evolution of Russia's economic differences is very important for improving China-Russia economic development cooperation and formulating China-Russia economic development plans. The Russian Federation is one of the most important participating countries in the "the Belt and Road" and "the economic corridor of China, Mongolia and Russia" initiatives. Under the background of "the Belt and Road" and "the economic corridor of China, Mongolia and Russia" initiatives, this paper studies the evolution characteristics of temporal and spatial pattern of Russian economic differences since the 21st century. The paper first used the weighted variation coefficient, Theil coefficient and concentration index to analyze the Russian economic disequilibrium changes during 2002–2020. Then combined with the regional economic grade index, this paper measured the economic grades of 83 Russian federal subjects, comparing the economic differences from the level of federal subjects during 2002–2020. Finally, the evolution characteristics of the temporal and spatial pattern of Russian economic differences were discussed by using the global trend analysis tool and spatial autocorrelation model. In the theory, this paper can reveal the spatial economic development process and spatial economic development regular pattern in the Russian Federation. It can explore the spatial economic development characteristics and spatial economic development model in the Russian Federation. It can summarize the functional positioning and industrial division of cities in the Russian Federation. It can provide the basis and conditions of bilateral and multilateral economic cooperation between Russia's neighboring countries and Russia. It also can clarify the bilateral and multilateral development patterns and problems between Russia's neighboring countries and Russia. These all have important theoretical significances for deepening the discipline systems of Economic Geography, Geo-economics and Regional Economics. In the practice, this paper can accelerate the connection between the "the economic corridor of China, Mongolia and Russia" initiative and the "trans-Eurasian Continental Bridge" initiative, promoting bilateral comprehensive cooperation and win-win development between Russia and China. It can help to clarify the complementary points of bilateral cooperation between China and Russia. It can provide a scientific reference for regional development planning, economic optimization layout, energy and resource development and infrastructure construction in the adjacent areas of China and Russia in the future. It can provide suggestions for adjusting economic cooperation field and expanding the investment scale in the border cities of China and Russia in the future. It can provide policy implications for determining the cooperation direction of border trade, transportation facilities, border tourism, border cooperation zone and ecological environment protection of China and Russia in the future. It also can provide scientific basis for the construction layout and economic cooperation along the economic corridor of China, Mongolia and Russia. These all have very important and urgent practical significance.

## Materials and method

### Study area

In 2020, the Russian Federation has a land area of 17.13 million square kilometers, a population of 146 million people, a gross regional product of 94.8 trillion rubles, a per capita gross

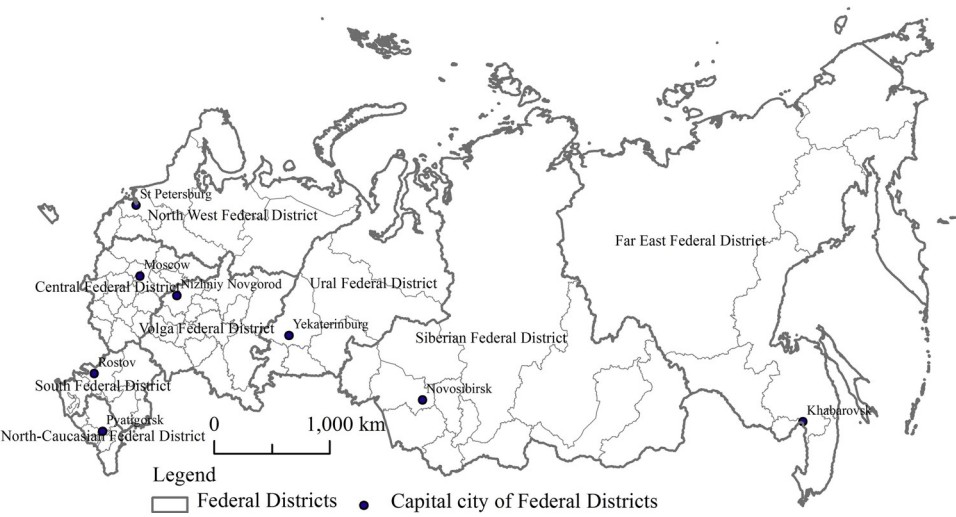

**Fig 1. Sketch map of the study area of the Russian Federation.**

regional product of 648800 rubles, a fixed capital investment of 20.12 trillion rubles, an average per capita money income (monthly) of 35700 rubles, and an average per capita money expenditure (monthly) of 27200 rubles. As shown in Fig 1, the Russian Federation has eight federal districts, including the Central Federal District, Ural Federal District, Volga Federal District, North West Federal District, South Federal District, North-Caucasian Federal District, Siberian Federal District, and Far East Federal District. It has eighty-five federal subjects, including forty-six regions, twenty-two republics, nine territories, four autonomous areas, three municipality cities, and one autonomous oblast. Due to the fact that the Sevastopol city and the Republic of Crimea are in dispute with the Ukrainian State, they are not selected as the objects of this study. So the eighty-three federal subjects are selected as the research objects of this paper.

## Research methods and data sources

**Economic development difference index.** This paper uses the weighted variation coefficient ($CV$), Theil coefficient ($T$), concentration index ($C$) to measure the economic development differences of the Russian Federation during 2002–2020. $CV$, $T$ and $C$ are used to measure the Russia's economic space differences, reflecting the Russia's economic spatial polarization degree [9, 10]. $CV$ reflects the dispersion degree of the economic development level from the perspective of the standard deviation. It is calculated by the ratio of the absolute difference to the average value. $T$ characterizes the overall economic space differences in the Russian Federation, exploring the impact of economic differences among federal subjects on the changes of Russia's overall differences. $C$ reflects the agglomeration degree of Russian economic factors in various federal subjects. The combination of the three indexes ($CV$, $T$, $C$) could make up for the errors of a single index. The $CV$, $T$ and $C$ are computed as follows:

$$CV = \frac{1}{\bar{x}}\sqrt{\sum_{i=1}^{n}(x_i - \bar{x})^2 \times \frac{p_i}{p}} \qquad T = \sum_{i=1}^{n} m_i \log\frac{m_i}{n_i} \qquad C = \sum_{i=1}^{n}(g_i/g)^2 \qquad (1)$$

where $CV$ is the weighted coefficient of variation, $T$ is the Theil index, and $C$ is the concentration index. A higher value of $CV$, $T$ and $C$ results in a higher economic development difference degree of the Russian Federation. $X_i$ is the per capita GDP of $i$ federal subject, $P_i$ is the

population of $i$ federal subject, and $G_i$ is the GDP of $i$ federal subject. $\bar{x}$ is the average per capita GDP of all the federal subjects. $P$ is the population of the Russian Federation and $G$ is the GDP of the Russian Federation. $m_i$ is the proportion of the GDP of $i$ federal subject in the GDP of Russian Federation. $n_i$ is the proportion of the population of $i$ federal subject in the population of Russian Federation. $n$ is the number of Russian federal subjects.

**Regional economic grade index.** The regional economic grade index is used to comprehensively measure the economic strength and economic status of the Russian federal subjects, reflecting the economic differences of the Russian Federation. The indicators of population and GDP are selected to reflect the overall economic development level of each federal subjects. The indicator of fixed capital investment is used to study the amount of economic activities such as the construction and fixed assets purchase of each federal subjects. The indicator of economic fixed assets is used to discuss the ability of enterprises to produce economic benefits from their production and operation activities in each federal subjects. The indicator of retail trade turnover is selected to study the level of goods and services sold by federal subjects through public trading platforms [1, 2, 10]. The regional economic grade index is computed as follows:

$$K_{Pi} = P_i \left/ \frac{1}{n} \sum_{i=1}^{n} P_i \right. \qquad K_{ti} = K_{Pi} + K_{Ei} + K_{Ci} + K_{Ri} + K_{Ti} \qquad K_{ei} = (K_{ti})/5 \qquad (2)$$

where $K_{Pi}$ is the grade index of population. $K_{Ei}$ is the grade index of GDP. $K_{Ci}$ is the grade index of the fixed capital investment. $K_{Ri}$ is the grade index of the economic fixed assets. $K_{Ti}$ is the grade index of the retail trade turnover. $K_{Ei}$, $K_{Ci}$, $K_{Ri}$, $K_{Ti}$ are calculated as $K_{Pi}$. $K_{ti}$ is the comprehensive economic grade index of the Russian federal subjects. $K_{ei}$ is the average economic grade index of the Russian federal subjects. $n$ is the number of Russian federal subjects. According to the natural discontinuity classification method, the regional economic grades of the Russian federal subjects are divided into five classes: first class ($K_{ei}$ between 5.65~12.94), second class ($K_{ei}$ between 2.79~5.64), third class ($K_{ei}$ between 1.54~2.78), forth class ($K_{ei}$ between 0.65~1.53) and fifth class ($K_{ei}$ between 0.05~0.64).

**Global trend analysis tool.** Using the global trend analysis tool in ArcGIS, this paper studies the overall characteristics of the economic differences of each federal subjects in the whole of the Russian Federation [1, 2, 10]. Firstly, this paper draws the position of each federal subject on the X-dimensional plane and Y-dimensional plane. Then, it projects the per capita GDP value of each federal subject onto the X-Y orthogonal plane and Y-Z orthogonal plane respectively. Next, based on the scatter diagrams projected on the X-Y plane and Y-Z plane, this paper uses the second-order polynomial for spatial best fitting. Finally, from a macro perspective, this paper analyzes the overall change trend of East-West and North-South economic differences of the Russian Federation during 2002–2020. The X-axis represents the east-west direction of the whole territory of Russia (the arrow points to the East), and the Y-axis represents the north-south direction of the whole territory of Russia (the arrow points to the North). The height of each vertical line of the Z-axis represents the per capita GDP of each federal subject.

**Spatial autocorrelation model.** The spatial autocorrelation model is used to analyze the economic spatial agglomeration mode, economic correlation structure and economic differentiation characteristics of the adjacent subjects in the Russian Federation. Spatial autocorrelation refers to the correlation of the same kind of variables in different spatial positions. It is a measure of the aggregation degree of attribute values of spatial units. It could represent the spatial interaction, spatial diffusion and spatial dependence between variable data at a certain

location and variable data at other locations. It contains the global spatial autocorrelation and the local spatial autocorrelation [1, 2, 21].

Global spatial autocorrelation is used to study the overall situation of spatial correlation and difference degree of unit attribute values in adjacent areas in the whole study area. In this paper, Moran index $I$ is used to measure the degree of global spatial autocorrelation. The Moran index $I$ is computed as follows:

$$I = \frac{\sum\limits_{i=1}^{n}\sum\limits_{j=1}^{n}w_{ij}(x_i - \bar{x})(x_j - \bar{x})}{S^2\sum\limits_{i=1}^{n}\sum\limits_{j=1}^{n}w_{ij}} \qquad S^2 = \frac{1}{n}\sum\limits_{i=1}^{n}(x_i - \bar{x})^2 \qquad \bar{x} = \frac{1}{n}\sum\limits_{i=1}^{n}x_i \qquad (3)$$

where $n$ is the number of Russian federal subjects. $w_{ij}$ is the spatial weight matrix, whose spatial proximity is defined as 1 and non adjacency is defined as 0. $x_i$ and $x_j$ are the attribute values of $i$ and $j$ spatial units, and $\bar{x}$ is their average values. The value range of Moran index $I$ is [−1,1]. If $I$ is greater than 0, there is a significant positive correlation in space. The larger the value, the stronger the spatial agglomeration trend. If $I$ is less than 0, there is a significant negative correlation in space, and the smaller the value, the stronger the spatial differentiation trend. If $I$ is equal to 0, it means that the space is uncorrelated.

Local spatial autocorrelation is used to study the differences of regional economic space in local scope, explaining whether there was spatial clustering and other correlation between the attribute values of local units and their adjacent units. In this paper, *Getis-Ord $G_i^*$* is used to measure the degree of local spatial autocorrelation. It could describe the spatial difference pattern among cold spots and hot spots, exploring its pattern difference characteristics. The *Getis-Ord $G_i^*$* is computed as follows:

$$G_i^*(d) = \sum\limits_{j=1}^{n} w_{ij}(d)x_j \Big/ \sum\limits_{j=1}^{n} x_j \qquad (4)$$

where $w_{ij}(d)$ is the spatial weight matrix between $i$ and $j$. $x_j$ is the attribute value of $j$ spatial unit. $n$ is the number of Russian federal subjects. This paper standardizes the $G_i^*(d)$. $Z(G_i^*) = (G_i^* - E(G_i^*))/\sqrt{Var(G_i^*)}$, $E(G_i^*)$ is the mathematical expectation of $G_i^*$. $Var(G_i^*)$ is the coefficient of variation of $G_i^*$. If $Z(G_i^*)$ is greater than 0, this federal subject tends to become the hot spot high-value agglomeration area. If $Z(G_i^*)$ is less than 0, this federal subject tends to become the cold spot low-value agglomeration area.

**Data sources.** The data for each indicator (population, GDP, the per capita GDP, the fixed capital investment, the economic fixed assets, and the retail trade turnover of eighty-three federal subjects) of the Russian Federation and its federal subjects comes from the 《*ФЕДЕРАЛЬНАЯ СЛУЖБА ГОСУДАРСТВЕННОЙ СТАТИСТИКИ (Росстат) РОССИЯ в цифрах Краткий статистический сборник*》 published on the official website of the Russian Bureau of Statistics for the period 2003–2021.

## Results

### Economic development differences changes

*CV* is greater than 0.50, *T* is greater than 0.10, and *C* is greater than 0.05. There are great differences in Russia's economic development. The three economic development difference indexes of *CV*, *T* and *C* of the Russian Federation showed a trend of fluctuating decreasing from 2002 to 2020 (Table 1). At the same time, as shown in Fig 2, the Moran index $I$ of Russian GDP also

**Table 1. Economic development difference indicators of the Russian Federation during 2002–2020.**

|  | 2002 | 2003 | 2004 | 2005 | 2006 | 2007 | 2008 | 2009 | 2010 | 2011 | 2012 | 2013 | 2014 | 2015 | 2016 | 2017 | 2018 | 2019 | 2020 |
|---|---|---|---|---|---|---|---|---|---|---|---|---|---|---|---|---|---|---|---|
| *CV* | 1.046 | 1.003 | 0.986 | 1.075 | 1.165 | 1.139 | 1.010 | 1.015 | 0.851 | 0.846 | 0.848 | 0.839 | 0.839 | 0.821 | 0.802 | 0.774 | 0.828 | 0.893 | 0.829 |
| *T* | 0.188 | 0.180 | 0.182 | 0.221 | 0.217 | 0.208 | 0.180 | 0.170 | 0.153 | 0.149 | 0.148 | 0.145 | 0.151 | 0.145 | 0.143 | 0.136 | 0.150 | 0.171 | 0.152 |
| *C* | 0.071 | 0.071 | 0.072 | 0.073 | 0.082 | 0.084 | 0.082 | 0.084 | 0.074 | 0.074 | 0.073 | 0.070 | 0.071 | 0.071 | 0.067 | 0.066 | 0.069 | 0.072 | 0.069 |

shows a fluctuating decreasing trend. Russia's economy shows a trend of changing from significant positive correlation in strong agglomeration space to positive correlation in weak agglomeration space, and then to random distribution. The economic imbalance among the Russian subjects shows a certain weakening trend. Natural conditions, institutional mechanisms, policy guidance, industrial structure and globalization development are the important factors that affect the economic development differences of Russian Federation.

First, natural conditions and institutional mechanisms are the important factors that affect the economic development differences of the Russian Federation. The Russian Federation has vast territory, and its terrain generally shows the characteristics of "high south, low north, high east and low west" [5]. There exists a large economic development difference among the northern, southern, eastern and western parts of the Russian Federation. The spatial distribution of the natural resources is not balanced in the Russian Federation. Mineral resources such as oil, gas, metals and nonmetals are mostly concentrated in the Far East and Siberian Federation Districts. And these mineral resources are far away from the main transportation lines, densely populated areas and economic consumption centers of the Russian Federation [5]. The northern and eastern parts of the Russian Federation have bad weather, insufficient population and lost talent. Their aging population has aggravated the shortage of labor force. And the underdevelopment of transportation and communication facilities has seriously affected the circulation of economic elements. During the Soviet Union period, the government implemented the state-led economic system. In order to balance the national productivity distribution and solve the war needs, Soviet Union put its economic centers on the central and eastern regions [22]. It implemented the population movement policy of moving eastwards, northwards, and southwards [23]. At the same time, under the planned economic system, it divided eleven national basic economic zones with relatively reasonable labor division, special industrial characteristics, and certain self-sufficiency capacity. But after the disintegration of the Soviet Union, the centrally centralized country transformed into the federalist country. The central and local governments of the Russian Federation had fierce competition. The Russian Federation divided various types of federal subjects such as region, autonomous area, republic, territory, and municipality city. Different federal subjects had the disparate scale, unequal legal status,

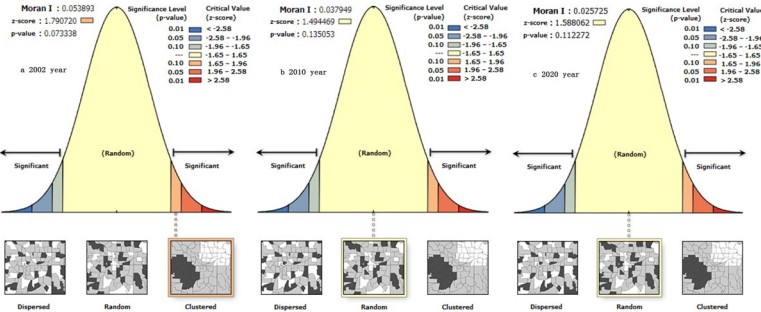

**Fig 2. Global autocorrelation index changes of GDP in Russian Federation from 2002 to 2020.**

and scattered financial power. The spatial friction of different federal subjects led to a series of new problems such as economy, trade and immigration. The international financial crisis in 1998 caused the economic chaos in the Russian Federation. Some of the federal subjects prevailed in endemism and they refused to pay taxes to the central government, forming the separate economic situations [24]. In addition, in the process of transition to the market economy, different regions formed different economic development models. For example, Moscow and some federal subjects formed a model of free economy. Nizhny Novgorod Region and some federal subjects formed a privatization economic model with western participation. Ulyanovsk Region and some federal subjects formed a gradual economic reform model. And Republic of Sakha(Yakutia) and some federal subjects formed an outward-oriented economic development model [24].

Second, policy guidance, industrial structure and globalization development are also the important factors that affect the economic development differences of the Russian Federation. After the disintegration of the Soviet Union in the 1990s, the Russian Federation had a shrinking economic space and weak inter-regional production connections. In order to solve the problems of political instability and regional separation, the policies and institutions were frequently adjusted at all departments in the Russian Federation. In 1996, the Russian Federation issued the "Primary Principles of Regional Policy in the Russian Federation" to solve the economic development problems in the federal districts. It tried to establish the free economic zones in the Republic of Kalmykia, Republic of Ingushetia, Republic of Altay, Sakhalin Region, and Jewish Autonomous Area to improve their economic development status. In the 2000–2004 years, the "Federal Goal Outline" was the policy guidance of the economic development in the Russian Federal Districts. In 2001, the Russian Federation promulgated the "Russian Federation Border Cooperation Concept" to build a win-win cooperation system between the border and its neighboring regions. At the same time, it provided financial subsidies and economic investment in the undeveloped regions such as the Far East Federal District, South Federal District, Baikal Region, Chechen Republic, Kaliningrad Region, and the special geopolitical areas such as Republic of Tatarstan, Republic of Bashkortostan, Sochi city to prevent their economic development imbalance. After 2004, the Russian government operated the legal regulation system of border cooperation. It promulgated policies such as "the Framework Convention on Border Cooperation between European Regional Organizations and Governments", "the Federal Law on the Coordination of International Relations and Foreign Economic Relations in the Russian Federation", "the Basic Law on State Regulation of Foreign Economic Activities", etc. These policies promoted the coordinated and balanced development of Russia's internal and external economy. In 2005–2007 years, the Russian Federation introduced the industrial cluster policies. And it encouraged the construction of the pharmaceutical industry in Saint Petersburg, the automobile manufacturing industry in Kaluga Region, the information technology and the pharmaceutical industries in Sverdlovsk Region, the aviation industry in Ulyanovsk Region, and the flax industry in Volgograd Region. In July 2005, the Russian Federation promulgated the "Law of the Special Economic Zone of the Russian Federation". It established three port logistics economic zones in the Khabarovsk Territory, Ulyanovsk Region and Murmansk Region. It established four industrial production economic zones in Republic of Tatarstan, Samara Region and Sverdlovsk Region. It established four technically-promoted economic zones in Moscow, Saint Petersburg, and Tomsk Region. It also established fourteen tourism and leisure economic zones in Republic of Altay, Republic of Buryatia, Irkutsk Region, Primorsky Territory and other federal subjects [25]. In 2005–2008 years, the Russian Federation successively carried out the five rounds of the federal subject's merger. The eighty-nine federal subjects of the Russian Federation had been reduced to eighty-three so as to avoid economic separatism. In 2008, the Russian Federation's financial crisis erupted. In

order to integrate economic resources, it enhanced the industrial competitiveness by improving the inter-regional labor specialization division. The Russian Federation introduced various anti-crisis policies such as stabilizing the budget, stimulating domestic demand and strengthening the basic resources of the banking industry, so as to narrow the gap of regional economic development and residents living quality. After 2008, the Russian Federation approved the "Strategic Concept for Long-term Economic and Social Development by 2020". It began to implement the economic strategy of multi polarization development. High-tech production clusters were constructed in the areas with high-levels of urbanization. Energy production and deep processing industrial clusters were established in the areas with low-levels of economic development. Tourism and leisure industrial clusters were developed in the areas with superior natural conditions. Large-scale logistics and production centers were constructed along the main transport lines. By the end of 2011, the eight federal districts had promulgated their economic and social development strategies. From the perspective of external environment, the globalization development had more and more influence on the economic development differences among the eight federal districts. The economic connections had been strengthened among the Central Federal District, North West Federal District and the European Union. The Central Asia and Southeast Asia were the economic backbones of the Ural Federal District and Western Siberian Federal District. The Asia-Pacific was the economic cooperation partner of the Far East Federal District and Baikal District. The Commonwealth of the Independent States (CIS) countries and China were the potential economic cooperation regions of the North-Caucasian Federal District and Eastern Siberian Federal District. In 2012–2014 years, the European Debt crisis and the Ukrainian crisis broke out in the Russian Federation. Europe, America and other countries cracked down on oil and gas industry, national defense and civilian products, etc in the Russian Federation. However, the Russian Federation implemented anti-sanctions measures such as freezing the assets of American enterprises and turning to cooperation with Eastern countries. At the same time, the Russian Federation implemented the policy of "regional ministry", retaining the affairs ministry of North-Caucasian Federal District and the developing ministry of Far East Federal District. While vigorously promoting the stability of the North-Caucasian Federal District and the development of Far East Federal District, the Russian Federation would further realize the balanced economic development strategy of multi polarization.

## Regional economic grade classification

In 2002, the $K_{ei}$ value of 74% federal subjects was lower than that of the average of the Russian Federation. In 2010, the $K_{ei}$ value of 71% of the federal subjects was lower than that of the average of the Russian Federation. And in 2020, the $K_{ei}$ value of 72% of the federal subjects was lower than that of the average of the Russian Federation. The first-class economic region is located in the Moscow City. The second-class economic regions are distributed in the Tumen Region, Moscow Region and Saint-Petersburg city. The third-class economic regions are located in the Krasnodar Territory, Khanty-Mansiysky Autonomous Area, Republic of Tatarstan, Sverdlovsk Region, Yamalo-Nenetsky Autonomous Area, Rostov Region, Republic of Bashkortostan, and Krasnoyarsk Territory. The forth-class economic regions are distributed in the Nizhny Novgorod Region, Samara Region, Chelyabinsk Region, Perm Territory, Novosibirsk Region and other twenty-one Russian federal subjects. The remaining Russian federal subjects belong to the fifth-class economic regions. The proportion of the number of the federal subjects in first-class, second-class, third-class, forth-class and fifth-class economic regions is 1.2%, 3.6%, 9.6%, 25.3% and 60.3% respectively. In 2020, the $K_{ei}$ value of Moscow City (12.94) is 217 times that of Chukotka Autonomous Area (0.06). Although the economic

imbalance among the federal subjects shows a weakening trend, the economic polarization between the developed prosperous regions and the stagnant backward regions is still severe.

Geographical location, energy resources and agglomeration effects are the important factors that affect the economic grade of the various federal subjects. The Russian Federation's economic resources are concentrated in the first-class, second-class and a few third-class federal subjects. More than one-third of Russia's GDP and economic fixed assets are concentrated in Moscow City, Tumen Region, Moscow Region and Saint-Petersburg city. More than one-forth of Russia's retail trade turnover is concentrated in Moscow City, Saint-Petersburg city and Moscow Region. Nearly one-second of Russia's fixed capital investment is concentrated in Moscow City, Tumen Region, Yamalo-Nenetsky Autonomous Area, Moscow Region, Khanty-Mansiysky Autonomous Area, Saint-Petersburg city and Republic of Tatarstan. Moscow City has the prominent agglomeration advantage that other federal subjects do not have. It has the decision-making right of the entire Russian economic development, and it implements the macroeconomic policies at the national level. It is the only huge economic cluster with a population of over ten million in the Russian Federation. It is also the Russian Federation's attraction center for industry, information, commodities, culture, finance and services, as well as a unique international metropolis supporting the Central Federal District. Tumen Region has the prominent geographical location advantage to develop the economic industries. It connects with the capitals of other federal subjects by the national roads and the Trans-Siberian railway. The good agricultural conditions make it possible to produce high yields of cereal crops in the south Tumen Region. Its machinery manufacturing industry provides various industrial products for the building, oil-gas chemical industry, transportation and other industries. The adequate power reserve and low natural gas prices make it become an excellent investment place. It is also planned as one of the high-tech development parks in the Russian Federation. The economic development of Khanty-Mansiysky Autonomous Area, Yamalo-Nenetsky Autonomous Area and Sverdlovsk Region depends on the advantages of energy resource. Khanty-Mansiysky Autonomous Area has the highest oil reserves, and it has the decisive right on the oil industry development and power production in the Russian Federation. Yamalo-Nenetsky Autonomous Area has the largest natural gas in the world. And its oil, natural gas and power industries are the leading industries that determine its economic prosperity. Sverdlovsk Region has the largest agricultural plantation in the Ural Federal District. Its rich mineral resources contribute to its heavy industrial development. It is also one of the scientific education centers in the Russian Federation. The Krasnodar Territory is rich in natural resources such as forests, minerals and fresh water. And its agricultural output such as grain is the highest in the Russian Federation. Republic of Tatarstan is a combination federal subject of the Central Russia and the Volga River. It has a well-developed transportation system, including the Trans-Siberian railway, the Volga river route, and the federal highway routes. It also has abundant petroleum field, the developed oil-gas extraction, chemical and mechanical manufacturing industries.

Spatially, the economic grade of the Russian federal subjects presents a significant spatial differentiation pattern (Fig 3). And it has a clear core-periphery pattern, which in generally displays a tendency of space attenuation from Ural Federal District and Volga Federal District to the North West Federal District, Siberian Federal District, and Far East Federal District. (1) Relying on the traditional petrochemical and machine manufacturing industries, Volga Federal District has made rapid progress in automobile and aircraft manufacturing, pharmaceutical, construction, energy and transportation industries. And it has outstanding advantages in industrial diversification, including the agricultural and industrial complexes. Relying on the superior location, rich resources, diversified economic structure and industrial division system of different federal subjects, Ural Federal District has large reserves of oil, gas, mineral, forest

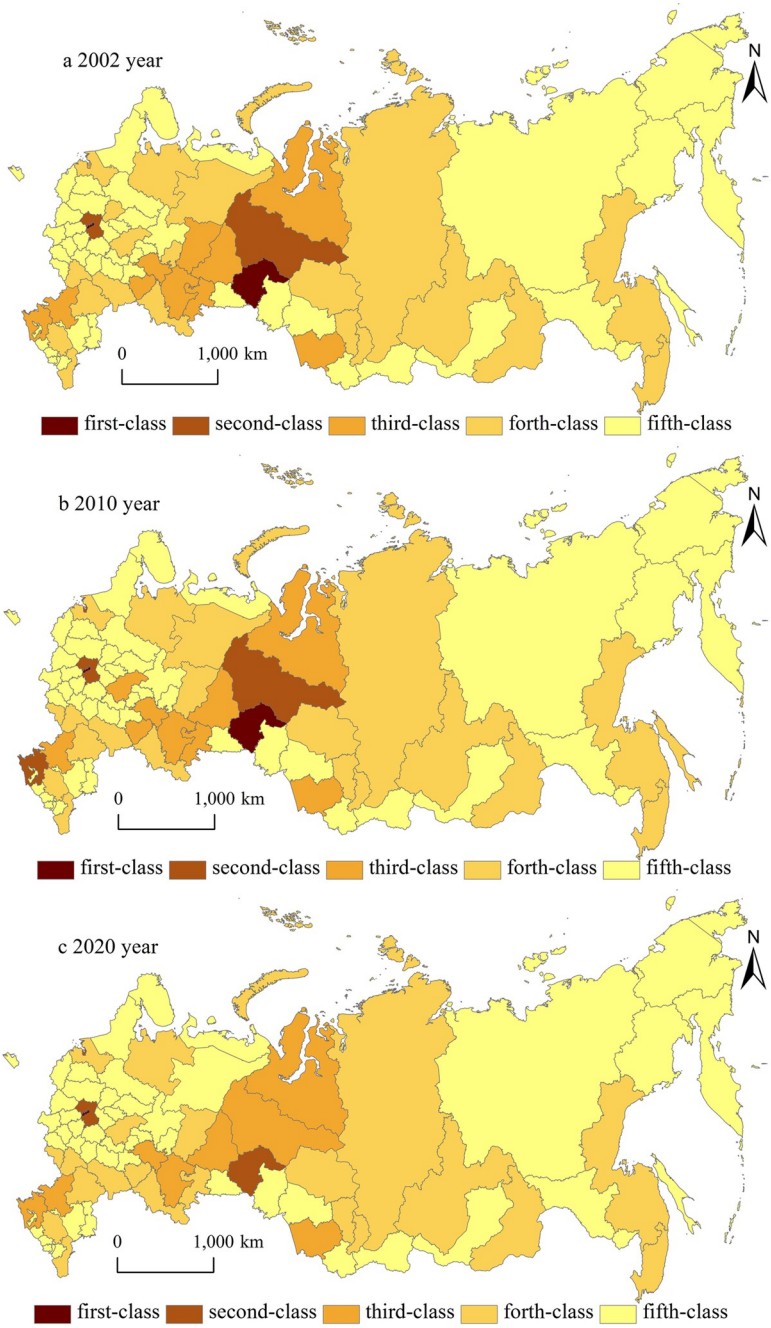

**Fig 3. Spatial pattern of economic grades in the Russian Federation from 2002 to 2020.**

and water resources to develop the metallurgy, energy, machinery, chemistry and other industries. And the oil and gas industry clusters have been formed on the Yamal Shelf of Karahai Bay and the Obskaya Guba. The combination area of Yamalo-Nenetsky Autonomous Area, Khanty-Mansiysky Autonomous Area, Sverdlovsk Region, and Tumen Region has formed the labor division clusters of different industries. The economic integration effect of this combination area has spread to the east and west sides of the Ural Federal District. (2) Within western Europe of the Russian Federation, the high economic grade regions are located in the Moscow

City and Moscow Region in the Central Federal District, Saint-Petersburg city in the Northwest Federal District, Krasnodar Territory and Rostov Region in the South Federal District, and Republic of Tatarstan in the Volga Federal District. In the Central Federal District, Moscow is the world's modern economic core and the Russian industrial leader. It has strong technological potential and modern logistics service system. It also has diversified innovative economic resources, and the accessible transportation network. The economic spillover effect of Moscow has brought the economic growth of its surrounding federal subjects. Other surrounding federal subjects become the Moscow's economic periphery areas. The economic development level of the Central Federal District presents a spatial pattern of "single pole-periphery nesting". In the Volga Federal District, the Republic of Tatarstan has formed the four main economic axes of port transportation, industrial production, technological innovation and sightseeing leisure relying on its northwest, northeast and southeast economic zones, the former- Kaamia, post-Kamia and the former Volga River's agricultural economic zones. It has a significant economic nuclear effect. The sub-core areas and edge areas are distributed in the southeast and northwest of the Republic of Tatarstan respectively. In the South Federal District, the Krasnodar Territory and Rostov Region are the economic growth poles. They have the agricultural, industrial, information and biotechnological complexes relying on the innovative economic platform. The sub-core areas and edge areas are distributed in the north and south of the Krasnodar Territory and Rostov Region respectively. (3) Within eastern Asia of the Russian Federation, the low economic grade areas are located in the south of Siberian Federal District and the north of the Far East Federal District, including the Chukotka Autonomous Area, Republic of Tyva, Jewish Autonomous Area, Republic of Altay, Kamchatka Territory and Magadan Region. These federal subjects have weak geographical advantages, poor transportation accessibility, complicated natural environments, and difficult resource exploration. And they have weak agglomeration effects, serious population loss, insufficient labor force, and imperfect market mechanisms. They also have single industrial structure, unsatisfactory investment environment, weak foreign investment attraction, insufficient development funds, and poor transportation infrastructure.

## Global economic trend analysis

From 2002 to 2020, the overall global economic trend of the Russian Federation represented by the per capita GDP shows some similarities. The overall economic differentiation characteristics of the Russian Federation are significant in east-west and north-south directions, which are closely related to the geographical location, resource conditions and economic agglomeration effect of the Russian federal subjects. The overall global economic trend of the Russian Federation shows the spatial characteristics of "High South, Low North" in the north-south direction. The overall global economic trend of the Russian Federation shows the spatial characteristics of " High West, Low East" in the east-west direction (Fig 4).

First, the economic development level in the southern part of the Russian Federation is stronger than that in the northern part of the Russian Federation. The northern part of the Russian Federation is in a high latitude area, which is not suitable for the economic development. It has bad climate, frozen soil, swamps and earthquakes. Most of the resource-based cities have difficulties in the industrial transformation and upgrading, and then they are abandoned as the uninhabited areas and "ghost cities". However, the northern part of the Russian Federation has great strategic significance in the resource reserves and ecological security. In order to solve the problem of regional economic imbalances, the Russian federal government expropriates the tax subsidies from the southern part to support the northern part of the Russian Federation. And financial transfer payments become the important income source for

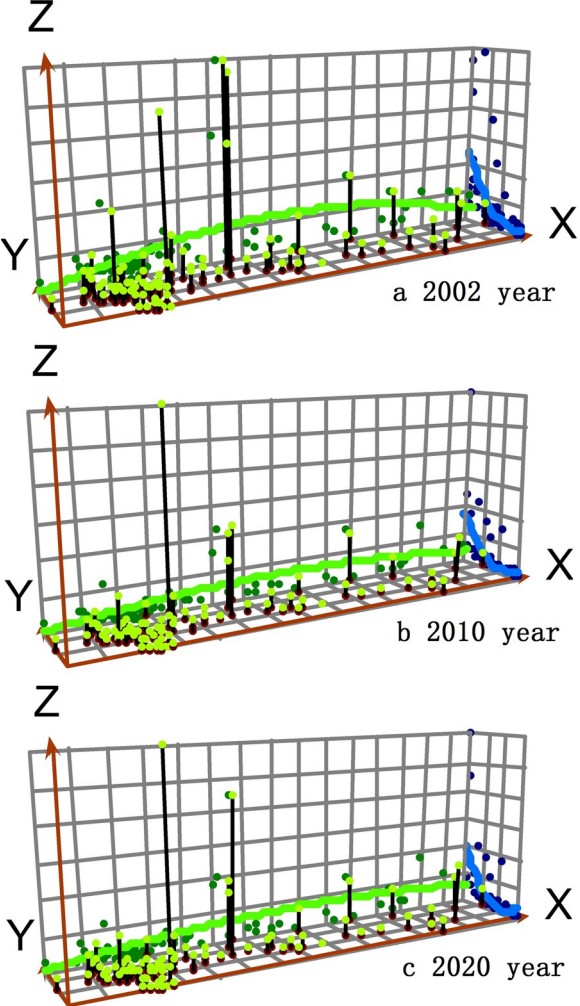

**Fig 4. Global trend analysis of the per capita GDP in the Russian Federation from 2002 to 2020.** Note: The X and Y axes are the east and north directions of the Russian Federation respectively. The Z axis is the per capita GDP of each federal subject. The green curve indicates the economic trend from east to west, and the blue curve indicates the economic trend from north to south.

the residents in the northern part of the Russian Federation. However, the northern part of the Russian Federation has the dysfunctional organization mechanism and weak economic growth. It's difficult to realize its high-speed economic operation in a short period of time. Once the Russian Federation's economic crisis erupts again, the economic development in the southern part of the Russian Federation will stagnate. And the economic vulnerability in the northern part of the Russian Federation will become more severe. The economic imbalance will be further enlarged between the northern part and the southern part of the Russian Federation.

Second, the economic development level in the western part of the Russian Federation is stronger than that in the eastern part of the Russian Federation. The Central, North West and other federal districts in the western part of the Russian Federation were the economic centers during Soviet Union period. They have the dense population, diverse industrial structure, strong economic strength, outstanding foreign trade, prominent scientific and technological innovation ability, and intensive transportation network. The development in the eastern part

of Russian Federation is mainly for the purpose of territorial security and national defense. The opening of the trans-Siberian railway in 1916 meant the real economic development in the eastern part of the Russian Federation. And the economic development of Siberian Federal District shows an attenuation pattern from Western Siberian District to Eastern Siberian District. The economic development of the Far East Federal District presents the spatial characteristics of "high south, low north". At present, the economic development in the western marginal part of the Russian Federation is decreasing, due to the oil and gas resources over-exploitation, reduced forest coverage, limited water resources, pollution of surface water, atmosphere, and land. With the economic rapid growth of the Asia-Pacific, president Putin realizes the great strategic significance to develop the Far East and Siberian Federal Districts. Although the Far East Federal District is sparsely populated, it has enormous potential to develop resources such as oil, gas, metals and forests. It has planned to exploit the southern urban agglomeration area and the island economic belt, and it has begun to construct the transportation, energy and other economic infrastructure. Siberian Federal District has planned to exploit three economic belts in its Arctic, northern and southern regions, relying on the innovative economic growth model integrating production, learning and research. The Far East and Siberian Federal Districts in the eastern part of the Russian Federation will become the important economic guarantee areas of the western part of the Russian Federation. And the expansion trend of economic difference is likely to be restrained between the eastern and western parts of the Russian Federation.

## Local economic hot-cold regions changes

The local spatial correlation index of Russian GDP was calculated based on *Getis-Ord Gi*[*] in ArcGIS. According to the natural discontinuity classification method, the pattern of cold and hot areas is divided into four levels (-1.56~-0.78 is cold spot region, -0.77~0.00 is sub-cold region, 0.01~1.25 is sub-hot region and 1.26~2.50 is hot spot region). From 2002 to 2020, there is no significant changes in the scope of the economic cold and hot regions in the Russian Federation. Only a small part of the hot regions are transformed into sub-hot regions, and a small part of the sub-cold regions are transformed into sub-hot regions. Spatially, similar to the pattern of economic grade, the economic hot and cold areas of the Russian Federation also presents a core-periphery pattern, which in generally displays a tendency of space attenuation from Ural Federal District to the western Russian Federation and eastern Russian Federation. And their regional correlation characteristics are significant (Fig 5). From 2002 to 2020, the Russian Federation is in the stage of economic factors gathering from marginal areas to core areas and the stage of dual structure deepening of core areas and marginal areas.

The economic hot regions are consistent with the high economic grade regions, which are mainly distributed in the Ural Federal District. The Ural Federal District is the largest producer of oil and gas in the Russian Federation. Its industrial production value accounts for more than 50% of its GDP. It is famous for the large industries complex including the machine manufacturing, fuel power, metal smelting, sawn wood pulp, chemical and other industries. It has two large agricultural areas in the northern non-black soil area and the southern Ural area. It has stable investment attraction ability, high-quality labor force and enormous innovative potential. It also has relatively developed transportation infrastructure, including twenty-eight airports. And it is the largest interregional trading partner of the Central Federal District. In addition, the economic hot region is also located in the Moscow City and Moscow Region in the Central Federal District. Moscow City and Moscow Region have relatively complete industrial departments, and the largest machine manufacturing industry in the Russian Federation. The military, power, chemicals, light industry, and building materials also play the decisive

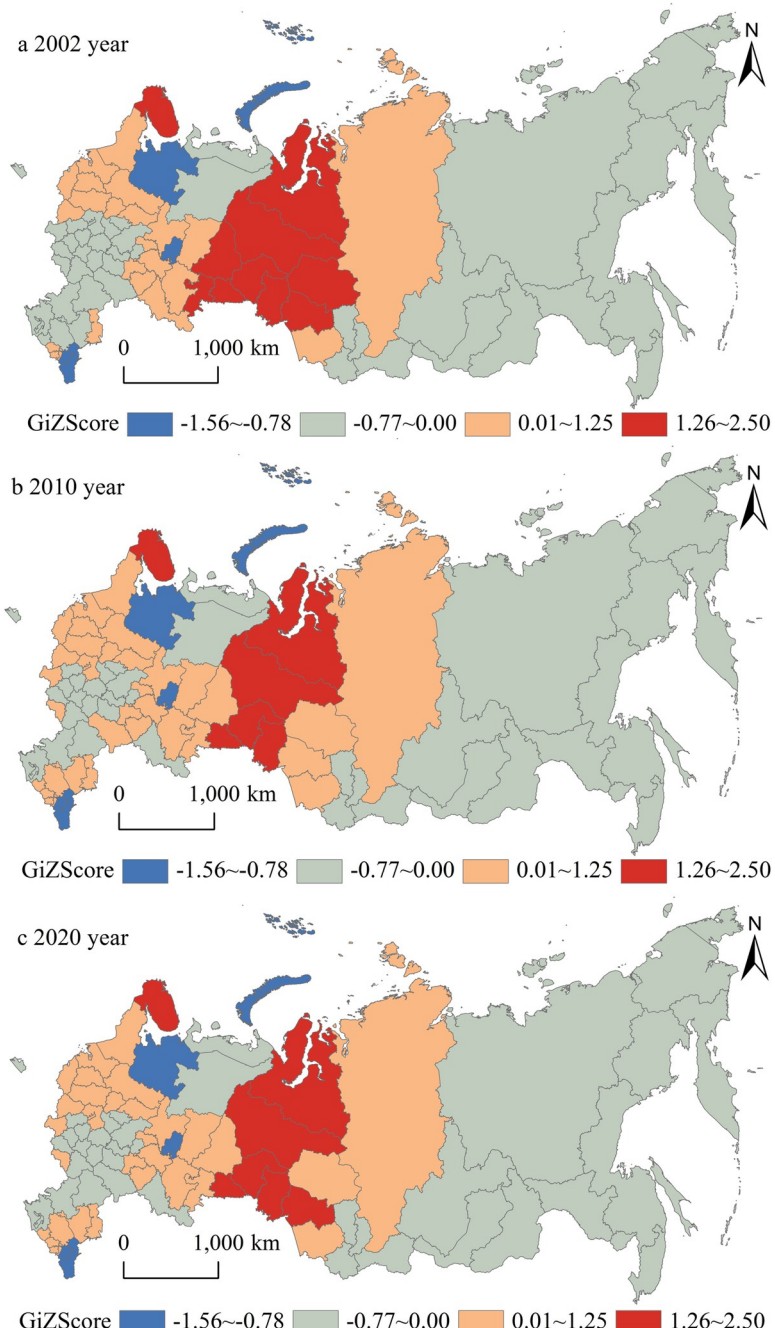

**Fig 5. Spatial pattern of local hot-cold regions changes in the Russian Federation from 2002 to 2020.**

industrial roles in the Moscow City and Moscow Region. The radial transport network of the Central Federal District has eleven international airports, eleven railways and fifteen highways, including the Trans-Siberian railway and the North-South transport corridor. The Central Federal District has concentrated about 80% technological potential and one-third of the small and medium enterprises (SMEs) of the Russian Federation.

The economic clod regions are consistent with the low economic grade regions, which are mainly located in the Far East Federal District, the east of Siberian Federal District, the north

of North West Federal District and the south of North-Caucasian Federal District. Far East Federal District and the east of Siberian Federal District have larger territory compared with other federal districts, but the population density is three persons per square kilometer. Due to the severe climate, the economic activities are mainly concentrated in the western Siberian Federal District. The eastern Siberian Federal District's ecological diversity is declining in these years. The Far East Federal District is far from the economic and political centers of the Russian Federation. It only has two railway lines, including the Trans-Siberian railway and Baikal-Amur railway lines. Some parts of eastern Siberian Federal District and Far East Federal District have low operating quality of the socio-economic system. Their population crisis has not been solved for a long time and their economic development is in a severe imbalance situation [26–29]. The economic growth of the eastern Siberian Federal District and Far East Federal District mainly depends on the input of labor and material capital, and the contribution rate of science and technology is relatively low [30]. At the same time, the local government lacks of the financial resources to maintain the normal operation of infrastructure, and it is difficult for residents to enjoy public services such as education, medical treatment, housing and so on. The two federal districts have poor corporate operating environments, imbalanced labor markets and severe capital losses. And their small businesses are prone to form economic monopolies. The agriculture and animal husbandry development of the north of North West Federal District is constrained by various factors. And there is a large differentiation between the supply and consumption of natural gas. Its transportation industry assets have a high level of depletion, and its nuclear facilities have caused large ecological danger. The south of North-Caucasian Federal District has a certain amount of mineral reserves, but these minerals have not been fully utilized. Its industrial structure is in a low development degree and it is strongly dependent on subsidies from the central federal government. Some of the federal subjects have unclear administrative boundaries and the land property rights in the North-Caucasian Federal District. The south of North-Caucasian Federal District's religious disputes, terrorism danger cannot be solved in the short term. It has high youth crimes rate such as drug abuse and alcohol abuse. The economic development pattern of the entire North-Caucasian Federal District presents low level homogenization characteristic. At present, it is difficult to guarantee the stability of the socio-political situation in the North-Caucasian Federal District.

## Discussion

The Russian Federation has vast territory and a large number of federal subjects. It has uneven population distribution and complicated ethnic issues. Under the background of two rounds of economic crisis in 2008 and 2014, the unbalanced regional economic development was not helpful to the political, economic, and social integration of the entire Russian Federation. At present, the Russian Federation has gradually got rid of the economic recession. However, its economic development still suffers the sanctions of European and American economies. It still has a long way to achieve balanced regional economic and social development.

First, the Russian Federation needs reform the political systems and economic systems. It needs to establish long-term and short-term economic and social development strategies at different scales, such as the Russian Federation, the Federal Districts, the Federal Subjects, the towns and the rural areas. It needs to continue to deepen the innovation development strategy of the "Russian 2020 Development Strategy" and the economic modernization plan of the "Basic Principles of Anti-crisis Action in 2010", relying on its domestic resources and intelligence advantages. Its economic development model should gradually transit from an energy resource export type to an innovative type. It needs to focus on promoting the development of the manufacturing industry, improving the business environment and enhancing the

investment attraction. It should implement a new round of financial tax reform, optimizing the structure of state-owned assets. It should guarantee the SMEs to develop the real economy, limiting the inflation rate. It also should promote employment rate, increasing the actual income of residents. It needs to establish a unified social and economic space and it should implement a relatively equitable distribution system by effective production distribution and reasonable labor division.

Second, the Russian Federation needs to implement unbalanced economic development strategies concerning to local natural conditions. The federal subjects in the western part of the Russian Federation continue to develop its traditional competitive advantages in energy resources. These federal subjects should build an energy price mechanism that is keeping with the international standards. And they should develop their potential advantages, increasing investment in high-tech fields. On the basis of realizing industrialization, they should upgrade their industrial structures. The federal subjects in the eastern part of the Russian Federation continue to focus on the economic development in resource-rich areas. These federal subjects should speed up the construction of energy transportation networking, improving the electricity infrastructure. They should apply high-new technology to create a high return rate on the economy, so as to absorb the return of labor and capital assets. They need to implement the diversified industrial development modes, improving the single structure of traditional energy resources.

Third, the Russian Federation should actively participate in the cooperation of multilateral economies in Asia-Pacific. It should improve its opening-up policies, participating in China's "the Belt and Road", "China- Mongolia-Russia Economic Corridor," and "Changchun-Jilin-Tumenjiang Development and Opening Pilot Zone" and other initiatives. With the advantages of geographical proximity and resources complementary, it should build a free trade zone in the Tumen River Delta to increase its participating efficiency in the international labor division. It should promulgate some preferential policies to attract investment of China, Japan and Korea to participate in its energy bases development by building the oil and gas export channels in the Siberian and Far East Federal Districts. It should actively participate in some transport and energy projects such as China's high-speed rail items. And then it should broaden the diversified cooperation structures of modern agriculture, manufacturing, construction, transportation and tourism, etc.

Forth, the Russian Federation should improve the transnational population migration policies. Its increasingly aging population and labor shortage have become the important constraints in the economic development in the Far East Federal District and other regions. It must implement ultra-conventional preferential policies to achieve the economic recovery and long-term stability in these regions. Every federal districts needs to stabilize the local population, preventing the population flow from the eastern part to the western part of the Russian Federation. The Russian Federation should promulgate more preferential policies to attract population migration to the Siberian and Far East Federal Districts. It should abandon the conservative xenophobia. It should promulgate more open and flexible immigration policies, creating a favorable investment environment to attract high-quality talents abroad. And it should create several economic growth poles in the Siberian and Far East Federal Districts.

Fifth, in recent years, with the economic rapid growth of the Asia-Pacific, Russian Federation has approved the "Society and Economic Development Plan for the Far East and Baikal Region" and the "Development Plan for the Border Areas of the Far East Federal District and Baikal Region" during 2014–2015. It has great strategic significance to develop the Far East Federal District and the Baikal region. First, developing the Far East and Baikal region can cope with the economic sanctions from the European and American countries, breaking through their export blockade. Second, developing the Far East Federal District and Baikal

regions can guarantee the geopolitical security and prevent population loss of the Russian Federation. Third, developing the Far East Federal District and Baikal region can serve as a new economic growth point to solve the uneven economic development of the eastern and western areas in the Russian Federation. Forth, developing the Far East Federal District and Baikal regions meets China's "Belt and Road Initiative". The Far East Federal District and Baikal regions will become key areas for the opening of the Russian Federation to the Asia Pacific.

The Russian Federation is an important neighboring country of China. Under the background of "the Belt and Road" and "the economic corridor of China, Mongolia and Russia" initiatives, bilateral relations between China and Russia have reached a high level of cooperation. In 2020, the trade volume between China and Russia reached 107.8 billion dollars. China has become Russia's largest trading partner for many years. In the future, the economic linkage strength and its pattern between China and Russia will become important research ideas. From the perspective of people flow, based on the modified models of population geographical concentration and population quotient, we will study the overall situation of cross-border labor market, labor migration and mobility intensity, and their impact on the local employment market between China and Russia. And we will also study the quantity, structure and behavior characteristics of cross-border tourists, the source and destination of cross-border tourists, and the impact of tourism activities on local prices, consumption, housing, culture, etc., so as to analyze the characteristics and process of people flow. From the perspective of economic flow, combined with the revised models of urban flow, economic linkage strength and geo-economic relations, we will study the import and export commodity structure, trade flow and direction, trade structure differences at border ports between China and Russia. And we will also study the supply and demand potential, spatial distance and transportation cost, economic interaction and economic radiation intensity of cities, so as to explore the characteristics and process of economic flow. From the perspective of traffic flow, we will use the normalized modified accessibility coefficient to calculate the relative value and dynamic change of accessibility of cities between China and Russia. We will use the weighted travel time, economic potential and daily accessibility to calculate e the improvement degree of accessibility of cities between China and Russia. We will use the transportation connection strength model to study the strength of transportation connection function of cities between China and Russia, so as to study the characteristics and process of traffic flow. From the perspective of comprehensive flow, we will give weights to the matrices of people flow, economic flow and traffic flow. Through a series of algorithms to obtain the comprehensive flow matrix, we will calculate the spatial comprehensive linkage strength and evolution between China and Russia from the perspective of multi-dimensional factor flow, so as to summarize the characteristics and process of comprehensive flow.

## Conclusions

Based on the economic development difference index, regional economic grade index, global trend analysis tool and spatial autocorrelation model, this paper analyzes the temporal and spatial pattern evolution characteristics of Russian economic differences from 2002 to 2020. The conclusions are as following:

The Russian Federation's economy presents spatial differentiation patterns of core-periphery pattern, east-west differentiation pattern and south-north differentiation pattern. The overall economic differentiation characteristics of the Russian Federation are significant in the east-west and north-south directions relatively, which are closely related to the geographical location, resource conditions and economic agglomeration effect of the Russian federal subjects. The overall global economic trend of the Russian Federation shows the spatial

characteristics of "High South, Low North" in the north-south direction. The overall global economic trend of the Russian Federation shows the spatial characteristics of " High West, Low East" in the east-west direction. However, the scope of its internal economic cold and hot areas has not changed significantly. The economic hot regions are mainly distributed in the Ural Federal District, Moscow City and Moscow Region. The economic clod regions are located in the Far East Federal District, the east of Siberian Federal District, the north of North West Federal District and the south of North-Caucasian Federal District.

Russia's economy shows a trend of changing from significant positive correlation in strong agglomeration space to positive correlation in weak agglomeration space, and then to random distribution. The Russian Federation's economic resources are concentrated in the first-class federal subject (Moscow City), second-class federal subjects (Tumen Region, Moscow Region and Saint-Petersburg city) and a few third-class federal subjects (Yamalo-Nenetsky Autonomous Area, Khanty-Mansiysky Autonomous Area, Republic of Tatarstan, Krasnodar Territory, Sverdlovsk Region, etc). Although the economic imbalance among the federal subjects shows a weakening trend, the economic polarization between the developed prosperous regions and the stagnant backward regions is still severe. Spatially, the economic grade of the Russian federal subjects presents a significant spatial differentiation pattern. And it has a clear core-periphery pattern, which in generally displays a tendency of space attenuation from Ural Federal District and Volga Federal District to the North West Federal District, Siberian Federal District, and Far East Federal District. Within western Europe of the Russian Federation, the high economic grade regions are located in the Moscow City and Moscow Region in the Central Federal District, Saint-Petersburg city in the Northwest Federal District, Krasnodar Territory and Rostov Region in the South Federal District, and Republic of Tatarstan in the Volga Federal District. Within eastern Asia of the Russian Federation, the low economic grade areas are located in the south of Siberian Federal District and the north of the Far East Federal District, including the Chukotka Autonomous Area, Republic of Tyva, Jewish Autonomous Area, Republic of Altay, Kamchatka Territory and Magadan Region.

## Supporting information

**S1 Table. Per capita GDP of the Russian federal subjects, 2002–2020 (person / ruble).** (XLS)

## Author Contributions

**Conceptualization:** Nan-Chen Chu.

**Data curation:** Nan-Chen Chu.

**Formal analysis:** Nan-Chen Chu.

**Funding acquisition:** Nan-Chen Chu, Xiang-Li Wu.

**Investigation:** Nan-Chen Chu, Ping-Yu Zhang.

**Methodology:** Nan-Chen Chu.

**Project administration:** Nan-Chen Chu.

**Resources:** Nan-Chen Chu, Xiang-Li Wu.

**Software:** Nan-Chen Chu, Ping-Yu Zhang, Xiang-Li Wu.

**Supervision:** Nan-Chen Chu.

**Validation:** Nan-Chen Chu.

**Visualization:** Nan-Chen Chu, Ping-Yu Zhang, Xiang-Li Wu.

**Writing – original draft:** Nan-Chen Chu.

**Writing – review & editing:** Nan-Chen Chu.

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
