## [Decision Letter · Decision Letter 0]

13 May 2021

PONE-D-21-11554

Economic Development Level and Its Regional Differences Analysis in Russian Federation

PLOS ONE

Dear Dr. Nanchen Chu,

Thank you for submitting your manuscript to PLOS ONE. After careful consideration, we feel that it has merit but does not fully meet PLOS ONE’s publication criteria as it currently stands. Therefore, we invite you to submit a revised version of the manuscript that addresses the points raised during the review process.

We look forward to receiving your revised manuscript.

Kind regards,

László VASA, PhD

Academic Editor

PLOS ONE

Journal Requirements:

3. Please amend the manuscript submission data (via Edit Submission) to include author ZHANG Pingyu, WU Xiangli，LI He.

4.  We note that Figures 1 and 4 in your submission contain map images which may be copyrighted. All PLOS content is published under the Creative Commons Attribution License (CC BY 4.0), which means that the manuscript, images, and Supporting Information files will be freely available online, and any third party is permitted to access, download, copy, distribute, and use these materials in any way, even commercially, with proper attribution. For these reasons, we cannot publish previously copyrighted maps or satellite images created using proprietary data, such as Google software (Google Maps, Street View, and Earth). For more information, see our copyright guidelines: http://journals.plos.org/plosone/s/licenses-and-copyright.

4.1.    You may seek permission from the original copyright holder of Figures 1 and 4 to publish the content specifically under the CC BY 4.0 license. 

4.2.    If you are unable to obtain permission from the original copyright holder to publish these figures under the CC BY 4.0 license or if the copyright holder’s requirements are incompatible with the CC BY 4.0 license, please either i) remove the figure or ii) supply a replacement figure that complies with the CC BY 4.0 license. Please check copyright information on all replacement figures and update the figure caption with source information. If applicable, please specify in the figure caption text when a figure is similar but not identical to the original image and is therefore for illustrative purposes only.

Reviewers' comments:

Reviewer's Responses to Questions

**Comments to the Author**

1. Is the manuscript technically sound, and do the data support the conclusions?

Reviewer #1: Partly

Reviewer #2: Partly

2. Has the statistical analysis been performed appropriately and rigorously? 

Reviewer #1: Yes

Reviewer #2: Yes

3. Have the authors made all data underlying the findings in their manuscript fully available?

Reviewer #1: No

Reviewer #2: Yes

4. Is the manuscript presented in an intelligible fashion and written in standard English?

Reviewer #1: No

Reviewer #2: Yes

5. Review Comments to the Author

Reviewer #1: The authors selected an interesting and timely topic as the subject of their research.

Regarding the datasets used, I would strongly recommend using the latest available data too, so even shifting the period of investigation to 2008-2018, as at this level of scientific writing to research a period ending six years ago is just not acceptable.

In fact, no literature review was made. The overview of the literature in the introduction part is very short and sketchy. An independent, comprehensive, analytical and critical literature review chapter should be written where essential works of other international authors would be processed. In addition, in the current version, the authors used almost exclusively Chinese sources.

Methodology description: the selected toolset should be set in context: giving some examples which other authors used the same method for research with a similar topic.

The results are supported by the methodology used by the authors and the conclusions are based on the results. However, in the case of the conclusions, the authors should conclude some lessons, consequences, recommendations, instead of just making statements based on the results.

Some technical notes:

Row 8 "of eight federal districts and eighty-three federal subjects" - I think this is not the proper word for that what you wanted to express.

Figure 1 - no source indicated (I suppose the map wasn't created by the authors as it seems to be an official map)

Figure 2 - no sources indicated

Figure 4 - by the "a)" map, probably there is a misspelling by the first (light yellow) category: it indicates .00-.08 while by the other three categories (b, c, d) it is .01-.08; please check

Table 1 - indicating the source of the table isn't right, it should be indicated under the table directly.

Table 2 - no source indicated

Table 3 - no source indicated

Reviewer #2: The paper analyses regional economic development patterns in Russia. Although the topic is interesting, the paper has a number of problems need to be handled:

1. The motivation behind the paper is poor. Why it is interesting to analyse this topic and what exactly this paper adds to the existing scientific literature?

2. The current literature review is mainly based on Russian and Chinese literature. I would advise the author to use a more extensive list of international literature to show the richness of this topic. Many authors have written similar papers to other regions - comparison among these studies would be a real added value.

3. What are the main conclusions coming out of previous literature? These should be drawn at the end of the literature review section.

4. The methodology seems to be very simple at first sight. Some more established economic methods would be needed to analyse this topic in more detail. The limitations of the method is also missing.

5. Comparison of own results with previous literature would also be needed when presenting the results.

6. What would be the recommendation for policy makers in Russia? How to change their current policies in line of the results?

7. What about future research ideas?

6. PLOS authors have the option to publish the peer review history of their article (what does this mean?). If published, this will include your full peer review and any attached files.

Reviewer #1: No

Reviewer #2: No

---

## [Author Response · Author response to Decision Letter 0]

9 Dec 2021

Thank you for your letter and for the reviewers' comments concerning our manuscript. These comments are very helpful for revising and improving our paper. We have studied comments carefully and have made corrections which we hope meet with approval.

Reviewer #1comments1: 

Regarding the datasets used, I would strongly recommend using the latest available data too, so even shifting the period of investigation to 2008-2018, as at this level of scientific writing to research a period ending six years ago is just not acceptable.

Response: Thanks for your valuable comments and suggestions. Through the official website of the Russian Bureau of Statistics, we have updated the data in this paper to the 2020 year. The data for each indicator (population, GDP, the per capita GDP, the fixed capital investment, the economic fixed assets, and the retail trade turnover of eighty-three federal subjects) of the Russian Federation and its federal subjects comes from the《ФЕДЕРАЛЬНАЯ СЛУЖБА ГОСУДАРСТВЕННОЙ СТАТИСТИКИ (Росстат) РОССИЯ в цифрах Краткий статистический сборник》published on the official website of the Russian Bureau of Statistics for the period 2003-2021. 

Thanks!

Reviewer #1comments2: 

In fact, no literature review was made. The overview of the literature in the introduction part is very short and sketchy. An independent, comprehensive, analytical and critical literature review chapter should be written where essential works of other international authors would be processed. In addition, in the current version, the authors used almost exclusively Chinese sources.

Response: Thanks for your valuable comments and suggestions. We have searched the international literatures on the Russian economic research. Then we write an independent, comprehensive, analytical and critical literature review chapter in the introduction section. The details are as follows:

The Russian Federation is an important neighboring country of China. Researches on Russia's economic development focus on the economic development situation, economic development differences, economic macroeconomic pattern, economic development characteristics, economic development trends, industrial development and reindustrialization of specific federal districts and federal subjects. Fedorov combined with Gini coefficient, GE index, ER index, Wolfson index and other indexes to study the polarization trend of Russian economic development from 1990 to 1999 [9]. Based on the hypothesis of spatial equilibrium and agglomeration economy, Kolomak believed that the Russian Federation had high spatial heterogeneity of economic activities, showing the agglomeration development. And the agglomeration speed in the west Europe was stronger than that in the east Asia [10]. Vertakova pointed out that the asymmetry of Russia's economic development was gradually weakening, but the economic imbalance degree was still high in the Russian Federation [11]. Granberg diagnosed the regionalization of the efficiency of Russia's economic anti-crisis planning, and he discussed the possible scenarios and future trends of Russia's economic recovery [12]. Taking the Siberian Federal District and the Far East Federal District as examples, Seliverstov compared competitive potential of labor and investment resources in economic development [13]. And from the efficiency of mineral and raw material development projects, Glazyrina studied the long-term economic benefits of cross-border cooperation between Russia and China [14]. Kuleshov discussed the development direction of reindustrialization planning in Novosibirsk, proposing the most competitive reindustrialization strategic measures with scientific innovation, engineering and manufacturing [15]. Kuz’minov studied the economic problems and social impact of the wood industry system in Kostroma Region, proposing its strategic countermeasures to adapt to the economic recession in 2009 [16]. The main conclusions drawn from the previous literatures were as followed. Since the period of economic transition, the uneven spatial allocation of industrial activities aggravated the polarization of regional economic and social development in the Russian Federation. And the economic differences among various regions had been increasing both qualitatively and quantitatively in the Russian Federation. This phenomenon had seriously restricted Russia's market reform and economic growth. The proportion of primary, secondary and tertiary industries was unbalanced in the Russian Federation, showing that the heavy industry was too heavy, the light industry was too light, the agriculture and the modern service industry fell behind for a long time. In addition, Russia's scientific and technological contribution rate was low, and Russia's economic development still depended on the labor and material capital investment. The economic growth rate of different Russian federal subjects was also unbalanced, which was reflected in the contraction of economic living space. In the future, the development of the Far East Federal District and the North-Caucasian Federal District will play a important role in promoting the regional economic balance in the Russian Federation [9-16].

Under the background of the rapid development of economic globalization, bilateral relations between China and Russia have reached a high level of cooperation. In 1996, China and Russia established a strategic cooperative partnership. In 2001, China and Russia signed the Sino-Russian treaty of good neighborliness, friendship and cooperation. In 2013, China and Russia established a new win-win cooperation relationship under the background of "the Belt and Road" initiative. The trade intensity, import and export volume between China and Russia had steadily increasing. In 2020, the trade volume between China and Russia reached 107.8 billion dollars. China has become Russia's largest trading partner for many years. The Russian Federation has formed a primary product export structure dominated by energy minerals to China [5, 17]. China has also formed a higher product export structure of machinery manufacturing, textile clothing and metal products to Russia [17-18]. At the same time, the traditional commodity services have expanded to the science-technology, transportation, tourism, military, environmental protection, energy and other fields in the Xinjiang-Western Siberia Federation district, Northeast China-Far East Federation district, and Northeast China-Siberian Federation district [18-20]. But unfortunately, the Corona Virus Disease 2019 (COVID-19) affected the overall economic activities between China and Russia. What should China and Russia do in the epidemic prevention and economic trade cooperation? The new topics are put forward for the scholars. Therefore, studying the temporal and spatial pattern evolution of Russia's economic differences is very important for improving China-Russia economic development cooperation and formulating China-Russia economic development plans. The Russian Federation is one of the most important participating countries in the "the Belt and Road" and "the economic corridor of China, Mongolia and Russia" initiatives. Under the background of "the Belt and Road" and "the economic corridor of China, Mongolia and Russia" initiatives, this paper studies the evolution characteristics of temporal and spatial pattern of Russian economic differences since the 21st century. The paper first used the weighted variation coefficient, Theil coefficient and concentration index to analyze the Russian economic disequilibrium changes during 2002-2020. Then combined with the regional economic grade index, this paper measured the economic grades of 83 Russian federal subjects, comparing the economic differences from the level of federal subjects during 2002-2020. Finally, the evolution characteristics of the temporal and spatial pattern of Russian economic differences were discussed by using the global trend analysis tool and spatial autocorrelation model. In the theory, this paper can reveal the spatial economic development process and spatial economic development regular pattern in the Russian Federation. It can explore the spatial economic development characteristics and spatial economic development model in the Russian Federation. It can summarize the functional positioning and industrial division of cities in the Russian Federation. It can provide the basis and conditions of bilateral and multilateral economic cooperation between Russia's neighboring countries and Russia. It also can clarify the bilateral and multilateral development patterns and problems between Russia's neighboring countries and Russia. These all have important theoretical significances for deepening the discipline systems of Economic Geography, Geo-economics and Regional Economics. In the practice, this paper can accelerate the connection between the "the economic corridor of China, Mongolia and Russia" initiative and the "trans-Eurasian Continental Bridge" initiative, promoting bilateral comprehensive cooperation and win-win development between Russia and China. It can help to clarify the complementary points of bilateral cooperation between China and Russia. It can provide a scientific reference for regional development planning, economic optimization layout, energy and resource development and infrastructure construction in the adjacent areas of China and Russia in the future. It can provide suggestions for adjusting economic cooperation field and expanding the investment scale in the border cities of China and Russia in the future. It can provide policy implications for determining the cooperation direction of border trade, transportation facilities, border tourism, border cooperation zone and ecological environment protection of China and Russia in the future. It also can provide scientific basis for the construction layout and economic cooperation along the economic corridor of China, Mongolia and Russia. These all have very important and urgent practical significance.

Thanks!

Reviewer #1comments3: 

Methodology description: the selected toolset should be set in context: giving some examples which other authors used the same method for research with a similar topic.

Response: Thanks for your valuable comments and suggestions. By searching the relevant literatures on Russian economic development, we introduce the new research methods to measure the Russian economic development differences. We study the evolution characteristics of temporal and spatial pattern of Russian economic differences during 2002-2020 with four methods, i.e. economic development difference index, regional economic grade index, global trend analysis tool, and spatial autocorrelation model. At the same time, in the process of introducing the methods, we also mark the cited literature. The details are as follows:

Research methods and data sources

Economic development difference index

This paper uses the weighted variation coefficient (CV), Theil coefficient (T), concentration index (C) to measure the economic development differences of the Russian Federation during 2002-2020. CV, T and C are used to measure the Russia's economic space differences, reflecting the Russia's economic spatial polarization degree [9-10]. CV reflects the dispersion degree of the economic development level from the perspective of the standard deviation. It is calculated by the ratio of the absolute difference to the average value. T characterizes the overall economic space differences in the Russian Federation, exploring the impact of economic differences among federal subjects on the changes of Russia's overall differences. C reflects the agglomeration degree of Russian economic factors in various federal subjects. The combination of the three indexes (CV, T, C) could make up for the errors of a single index. 

where CV is the weighted coefficient of variation, T is the Theil index, and C is the concentration index. A higher value of CV, T and C results in a higher economic development difference degree of the Russian Federation. Xi is the per capita GDP of i federal subject, Pi is the population of i federal subject, and Gi is the GDP of i federal subject. is the average per capita GDP of all the federal subjects. P is the population of the Russian Federation and G is the GDP of the Russian Federation. mi is the proportion of the GDP of i federal subject in the GDP of Russian Federation. ni is the proportion of the population of i federal subject in the population of Russian Federation. n is the number of Russian federal subjects. 

Regional economic grade index

The regional economic grade index is used to comprehensively measure the economic strength and economic status of the Russian federal subjects, reflecting the economic differences of the Russian Federation. The indicators of population and GDP are selected to reflect the overall economic development level of each federal subjects. The indicator of fixed capital investment is used to study the amount of economic activities such as the construction and fixed assets purchase of each federal subjects. The indicator of economic fixed assets is used to discuss the ability of enterprises to produce economic benefits from their production and operation activities in each federal subjects. The indicator of retail trade turnover is selected to study the level of goods and services sold by federal subjects through public trading platforms[1-2,10].

where KPi is the grade index of population. KEi is the grade index of GDP. KCi is the grade index of the fixed capital investment. KRi is the grade index of the economic fixed assets. KTi is the grade index of the retail trade turnover. KEi, KCi, KRi, KTi are calculated as KPi. Kti is the comprehensive economic grade index of the Russian federal subjects. Kei is the average economic grade index of the Russian federal subjects. n is the number of Russian federal subjects. According to the natural discontinuity classification method, the regional economic grades of the Russian federal subjects are divided into five classes: first class (Kei between 5.65~12.94), second class (Kei between 2.79~5.64), third class (Kei between 1.54~2.78), forth class (Kei between 0.65~1.53) and fifth class (Kei between 0.05~0.64).

Global trend analysis tool

Using the global trend analysis tool in ArcGIS, this paper studies the overall characteristics of the economic differences of each federal subjects in the whole of the Russian Federation [1-2,10]. Firstly, this paper draws the position of each federal subject on the X-dimensional plane and Y-dimensional plane. Then, it projects the per capita GDP value of each federal subject onto the X-Y orthogonal plane and Y-Z orthogonal plane respectively. Next, based on the scatter diagrams projected on the X-Y plane and Y-Z plane, this paper uses the second-order polynomial for spatial best fitting. Finally, from a macro perspective, this paper analyzes the overall change trend of East-West and North-South economic differences of the Russian Federation during 2002-2020. The X-axis represents the east-west direction of the whole territory of Russia (the arrow points to the East), and the Y-axis represents the north-south direction of the whole territory of Russia (the arrow points to the North). The height of each vertical line of the Z-axis represents the per capita GDP of each federal subject.

Spatial autocorrelation model

The spatial autocorrelation model is used to analyze the economic spatial agglomeration mode, economic correlation structure and economic differentiation characteristics of the adjacent subjects in the Russian Federation. Spatial autocorrelation refers to the correlation of the same kind of variables in different spatial positions. It is a measure of the aggregation degree of attribute values of spatial units. It could represent the spatial interaction, spatial diffusion and spatial dependence between variable data at a certain location and variable data at other locations. It contains the global spatial autocorrelation and the local spatial autocorrelation[1-2,21]. 

Global spatial autocorrelation is used to study the overall situation of spatial correlation and difference degree of unit attribute values in adjacent areas in the whole study area. In this paper, Moran index I is used to measure the degree of global spatial autocorrelation.

Local spatial autocorrelation is used to study the differences of regional economic space in local scope, explaining whether there was spatial clustering and other correlation between the attribute values of local units and their adjacent units. In this paper, Getis-Ord Gi* is used to measure the degree of local spatial autocorrelation. It could describe the spatial difference pattern among cold spots and hot spots, exploring its pattern difference characteristics. 

Thanks!

Reviewer #1comments4: 

The results are supported by the methodology used by the authors and the conclusions are based on the results. However, in the case of the conclusions, the authors should conclude some lessons, consequences, recommendations, instead of just making statements based on the results.

Response: Thanks for your valuable comments and suggestions. Due to the update of data and the replacement of methods, we recalculate the Russian economic development differences. And we redraw the pictures about the temporal and spatial pattern of Russia's economic development. Based on the original manuscript, we raise many new lessons, consequences, and recommendations about the Russian economic development. Please see the revised version of manuscript for details. 

Thanks once more!

Reviewer #1comments5: 

Some technical notes:

Row 8 "of eight federal districts and eighty-three federal subjects" - I think this is not the proper word for that what you wanted to express.

Figure 1 - no source indicated (I suppose the map wasn't created by the authors as it seems to be an official map)

Figure 2 - no sources indicated

Figure 4 - by the "a)" map, probably there is a misspelling by the first (light yellow) category: it indicates .00-.08 while by the other three categories (b, c, d) it is .01-.08; please check

Table 1 - indicating the source of the table isn't right, it should be indicated under the table directly.

Table 2 - no source indicated

Table 3 - no source indicated

Response: Thanks for your valuable comments and suggestions. Due to the update of data and the replacement of methods, we redraw the relevant tables and figures. Especially in the figures, the source of the figures has been indicated. Besides, by searching the relevant literatures about the Russian Federation, "federal districts" "federal subjects" are the proper words for scholars to express indeed. The details are as follows:

Note: this drawing is based on the standard map of the standard map service system of the Ministry of natural resources of China (drawing review No. GS (2016) 2276), and the base map is not modified. 

Fig 1. Sketch map of the study area of the Russian Federation

Fig 3. Spatial pattern of economic grades in the Russian Federation from 2002 to 2020

Fig 4. Global trend analysis of the per capita GDP in the Russian Federation from 2002 to 2020

Thanks once more!

Reviewer #2comments1: 

The motivation behind the paper is poor. Why it is interesting to analyse this topic and what exactly this paper adds to the existing scientific literature?

Response: Thanks for your valuable comments and suggestions. We have written the purpose and significance of this paper in the introduction again. The details are as follows:

Under the background of the rapid development of economic globalization, bilateral relations between China and Russia have reached a high level of cooperation. In 1996, China and Russia established a strategic cooperative partnership. In 2001, China and Russia signed the Sino-Russian treaty of good neighborliness, friendship and cooperation. In 2013, China and Russia established a new win-win cooperation relationship under the background of "the Belt and Road" initiative. The trade intensity, import and export volume between China and Russia had steadily increasing. In 2020, the trade volume between China and Russia reached 107.8 billion dollars. China has become Russia's largest trading partner for many years. The Russian Federation has formed a primary product export structure dominated by energy minerals to China [5, 17]. China has also formed a higher product export structure of machinery manufacturing, textile clothing and metal products to Russia [17-18]. At the same time, the traditional commodity services have expanded to the science-technology, transportation, tourism, military, environmental protection, energy and other fields in the Xinjiang-Western Siberia Federation district, Northeast China-Far East Federation district, and Northeast China-Siberian Federation district [18-20]. But unfortunately, the Corona Virus Disease 2019 (COVID-19) affected the overall economic activities between China and Russia. What should China and Russia do in the epidemic prevention and economic trade cooperation? The new topics are put forward for the scholars. Therefore, studying the temporal and spatial pattern evolution of Russia's economic differences is very important for improving China-Russia economic development cooperation and formulating China-Russia economic development plans. The Russian Federation is one of the most important participating countries in the "the Belt and Road" and "the economic corridor of China, Mongolia and Russia" initiatives. Under the background of "the Belt and Road" and "the economic corridor of China, Mongolia and Russia" initiatives, this paper studies the evolution characteristics of temporal and spatial pattern of Russian economic differences since the 21st century. The paper first used the weighted variation coefficient, Theil coefficient and concentration index to analyze the Russian economic disequilibrium changes during 2002-2020. Then combined with the regional economic grade index, this paper measured the economic grades of 83 Russian federal subjects, comparing the economic differences from the level of federal subjects during 2002-2020. Finally, the evolution characteristics of the temporal and spatial pattern of Russian economic differences were discussed by using the global trend analysis tool and spatial autocorrelation model. In the theory, this paper can reveal the spatial economic development process and spatial economic development regular pattern in the Russian Federation. It can explore the spatial economic development characteristics and spatial economic development model in the Russian Federation. It can summarize the functional positioning and industrial division of cities in the Russian Federation. It can provide the basis and conditions of bilateral and multilateral economic cooperation between Russia's neighboring countries and Russia. It also can clarify the bilateral and multilateral development patterns and problems between Russia's neighboring countries and Russia. These all have important theoretical significances for deepening the discipline systems of Economic Geography, Geo-economics and Regional Economics. In the practice, this paper can accelerate the connection between the "the economic corridor of China, Mongolia and Russia" initiative and the "trans-Eurasian Continental Bridge" initiative, promoting bilateral comprehensive cooperation and win-win development between Russia and China. It can help to clarify the complementary points of bilateral cooperation between China and Russia. It can provide a scientific reference for regional development planning, economic optimization layout, energy and resource development and infrastructure construction in the adjacent areas of China and Russia in the future. It can provide suggestions for adjusting economic cooperation field and expanding the investment scale in the border cities of China and Russia in the future. It can provide policy implications for determining the cooperation direction of border trade, transportation facilities, border tourism, border cooperation zone and ecological environment protection of China and Russia in the future. It also can provide scientific basis for the construction layout and economic cooperation along the economic corridor of China, Mongolia and Russia. These all have very important and urgent practical significance.

Thanks!

Reviewer #2comments2: 

The current literature review is mainly based on Russian and Chinese literature. I would advise the author to use a more extensive list of international literature to show the richness of this topic. Many authors have written similar papers to other regions - comparison among these studies would be a real added value.

Response: Thanks for your valuable comments and suggestions. We have searched the international literatures on the Russian economic research. Then we write an independent, comprehensive, analytical and critical literature review chapter in the introduction section. The details are as follows:

The Russian Federation is an important neighboring country of China. Researches on Russia's economic development focus on the economic development situation, economic development differences, economic macroeconomic pattern, economic development characteristics, economic development trends, industrial development and reindustrialization of specific federal districts and federal subjects. Fedorov combined with Gini coefficient, GE index, ER index, Wolfson index and other indexes to study the polarization trend of Russian economic development from 1990 to 1999 [9]. Based on the hypothesis of spatial equilibrium and agglomeration economy, Kolomak believed that the Russian Federation had high spatial heterogeneity of economic activities, showing the agglomeration development. And the agglomeration speed in the west Europe was stronger than that in the east Asia [10]. Vertakova pointed out that the asymmetry of Russia's economic development was gradually weakening, but the economic imbalance degree was still high in the Russian Federation [11]. Granberg diagnosed the regionalization of the efficiency of Russia's economic anti-crisis planning, and he discussed the possible scenarios and future trends of Russia's economic recovery [12]. Taking the Siberian Federal District and the Far East Federal District as examples, Seliverstov compared competitive potential of labor and investment resources in economic development [13]. And from the efficiency of mineral and raw material development projects, Glazyrina studied the long-term economic benefits of cross-border cooperation between Russia and China [14]. Kuleshov discussed the development direction of reindustrialization planning in Novosibirsk, proposing the most competitive reindustrialization strategic measures with scientific innovation, engineering and manufacturing [15]. Kuz’minov studied the economic problems and social impact of the wood industry system in Kostroma Region, proposing its strategic countermeasures to adapt to the economic recession in 2009 [16]. The main conclusions drawn from the previous literatures were as followed. Since the period of economic transition, the uneven spatial allocation of industrial activities aggravated the polarization of regional economic and social development in the Russian Federation. And the economic differences among various regions had been increasing both qualitatively and quantitatively in the Russian Federation. This phenomenon had seriously restricted Russia's market reform and economic growth. The proportion of primary, secondary and tertiary industries was unbalanced in the Russian Federation, showing that the heavy industry was too heavy, the light industry was too light, the agriculture and the modern service industry fell behind for a long time. In addition, Russia's scientific and technological contribution rate was low, and Russia's economic development still depended on the labor and material capital investment. The economic growth rate of different Russian federal subjects was also unbalanced, which was reflected in the contraction of economic living space. In the future, the development of the Far East Federal District and the North-Caucasian Federal District will play a important role in promoting the regional economic balance in the Russian Federation [9-16].

Under the background of the rapid development of economic globalization, bilateral relations between China and Russia have reached a high level of cooperation. In 1996, China and Russia established a strategic cooperative partnership. In 2001, China and Russia signed the Sino-Russian treaty of good neighborliness, friendship and cooperation. In 2013, China and Russia established a new win-win cooperation relationship under the background of "the Belt and Road" initiative. The trade intensity, import and export volume between China and Russia had steadily increasing. In 2020, the trade volume between China and Russia reached 107.8 billion dollars. China has become Russia's largest trading partner for many years. The Russian Federation has formed a primary product export structure dominated by energy minerals to China [5, 17]. China has also formed a higher product export structure of machinery manufacturing, textile clothing and metal products to Russia [17-18]. At the same time, the traditional commodity services have expanded to the science-technology, transportation, tourism, military, environmental protection, energy and other fields in the Xinjiang-Western Siberia Federation district, Northeast China-Far East Federation district, and Northeast China-Siberian Federation district [18-20]. But unfortunately, the Corona Virus Disease 2019 (COVID-19) affected the overall economic activities between China and Russia. What should China and Russia do in the epidemic prevention and economic trade cooperation? The new topics are put forward for the scholars. Therefore, studying the temporal and spatial pattern evolution of Russia's economic differences is very important for improving China-Russia economic development cooperation and formulating China-Russia economic development plans. The Russian Federation is one of the most important participating countries in the "the Belt and Road" and "the economic corridor of China, Mongolia and Russia" initiatives. Under the background of "the Belt and Road" and "the economic corridor of China, Mongolia and Russia" initiatives, this paper studies the evolution characteristics of temporal and spatial pattern of Russian economic differences since the 21st century. The paper first used the weighted variation coefficient, Theil coefficient and concentration index to analyze the Russian economic disequilibrium changes during 2002-2020. Then combined with the regional economic grade index, this paper measured the economic grades of 83 Russian federal subjects, comparing the economic differences from the level of federal subjects during 2002-2020. Finally, the evolution characteristics of the temporal and spatial pattern of Russian economic differences were discussed by using the global trend analysis tool and spatial autocorrelation model. In the theory, this paper can reveal the spatial economic development process and spatial economic development regular pattern in the Russian Federation. It can explore the spatial economic development characteristics and spatial economic development model in the Russian Federation. It can summarize the functional positioning and industrial division of cities in the Russian Federation. It can provide the basis and conditions of bilateral and multilateral economic cooperation between Russia's neighboring countries and Russia. It also can clarify the bilateral and multilateral development patterns and problems between Russia's neighboring countries and Russia. These all have important theoretical significances for deepening the discipline systems of Economic Geography, Geo-economics and Regional Economics. In the practice, this paper can accelerate the connection between the "the economic corridor of China, Mongolia and Russia" initiative and the "trans-Eurasian Continental Bridge" initiative, promoting bilateral comprehensive cooperation and win-win development between Russia and China. It can help to clarify the complementary points of bilateral cooperation between China and Russia. It can provide a scientific reference for regional development planning, economic optimization layout, energy and resource development and infrastructure construction in the adjacent areas of China and Russia in the future. It can provide suggestions for adjusting economic cooperation field and expanding the investment scale in the border cities of China and Russia in the future. It can provide policy implications for determining the cooperation direction of border trade, transportation facilities, border tourism, border cooperation zone and ecological environment protection of China and Russia in the future. It also can provide scientific basis for the construction layout and economic cooperation along the economic corridor of China, Mongolia and Russia. These all have very important and urgent practical significance.

Thanks!

Reviewer #2comments3: 

What are the main conclusions coming out of previous literature? These should be drawn at the end of the literature review section.

Response: Thanks for your valuable comments and suggestions. We have written the main conclusions coming out of previous literatures at the end of the literature review section in the introduction. The details are as follows:

The Russian Federation is an important neighboring country of China. Researches on Russia's economic development focus on the economic development situation, economic development differences, economic macroeconomic pattern, economic development characteristics, economic development trends, industrial development and reindustrialization of specific federal districts and federal subjects. Fedorov combined with Gini coefficient, GE index, ER index, Wolfson index and other indexes to study the polarization trend of Russian economic development from 1990 to 1999 [9]. Based on the hypothesis of spatial equilibrium and agglomeration economy, Kolomak believed that the Russian Federation had high spatial heterogeneity of economic activities, showing the agglomeration development. And the agglomeration speed in the west Europe was stronger than that in the east Asia [10]. Vertakova pointed out that the asymmetry of Russia's economic development was gradually weakening, but the economic imbalance degree was still high in the Russian Federation [11]. Granberg diagnosed the regionalization of the efficiency of Russia's economic anti-crisis planning, and he discussed the possible scenarios and future trends of Russia's economic recovery [12]. Taking the Siberian Federal District and the Far East Federal District as examples, Seliverstov compared competitive potential of labor and investment resources in economic development [13]. And from the efficiency of mineral and raw material development projects, Glazyrina studied the long-term economic benefits of cross-border cooperation between Russia and China [14]. Kuleshov discussed the development direction of reindustrialization planning in Novosibirsk, proposing the most competitive reindustrialization strategic measures with scientific innovation, engineering and manufacturing [15]. Kuz’minov studied the economic problems and social impact of the wood industry system in Kostroma Region, proposing its strategic countermeasures to adapt to the economic recession in 2009 [16]. The main conclusions drawn from the previous literatures were as followed. Since the period of economic transition, the uneven spatial allocation of industrial activities aggravated the polarization of regional economic and social development in the Russian Federation. And the economic differences among various regions had been increasing both qualitatively and quantitatively in the Russian Federation. This phenomenon had seriously restricted Russia's market reform and economic growth. The proportion of primary, secondary and tertiary industries was unbalanced in the Russian Federation, showing that the heavy industry was too heavy, the light industry was too light, the agriculture and the modern service industry fell behind for a long time. In addition, Russia's scientific and technological contribution rate was low, and Russia's economic development still depended on the labor and material capital investment. The economic growth rate of different Russian federal subjects was also unbalanced, which was reflected in the contraction of economic living space. In the future, the development of the Far East Federal District and the North-Caucasian Federal District will play a important role in promoting the regional economic balance in the Russian Federation [9-16].

Thanks!

Reviewer #2comments4: 

The methodology seems to be very simple at first sight. Some more established economic methods would be needed to analyse this topic in more detail. The limitations of the method is also missing.

Response: Thanks for your valuable comments and suggestions. By searching the relevant literatures on Russian economic development, we introduce the new research methods to measure the Russian economic development differences. We study the evolution characteristics of temporal and spatial pattern of Russian economic differences during 2002-2020 with four methods, i.e. economic development difference index, regional economic grade index, global trend analysis tool, and spatial autocorrelation model. The details are as follows:

Research methods and data sources

Economic development difference index

This paper uses the weighted variation coefficient (CV), Theil coefficient (T), concentration index (C) to measure the economic development differences of the Russian Federation during 2002-2020. CV, T and C are used to measure the Russia's economic space differences, reflecting the Russia's economic spatial polarization degree [9-10]. CV reflects the dispersion degree of the economic development level from the perspective of the standard deviation. It is calculated by the ratio of the absolute difference to the average value. T characterizes the overall economic space differences in the Russian Federation, exploring the impact of economic differences among federal subjects on the changes of Russia's overall differences. C reflects the agglomeration degree of Russian economic factors in various federal subjects. The combination of the three indexes (CV, T, C) could make up for the errors of a single index. 

where CV is the weighted coefficient of variation, T is the Theil index, and C is the concentration index. A higher value of CV, T and C results in a higher economic development difference degree of the Russian Federation. Xi is the per capita GDP of i federal subject, Pi is the population of i federal subject, and Gi is the GDP of i federal subject. is the average per capita GDP of all the federal subjects. P is the population of the Russian Federation and G is the GDP of the Russian Federation. mi is the proportion of the GDP of i federal subject in the GDP of Russian Federation. ni is the proportion of the population of i federal subject in the population of Russian Federation. n is the number of Russian federal subjects. 

Regional economic grade index

The regional economic grade index is used to comprehensively measure the economic strength and economic status of the Russian federal subjects, reflecting the economic differences of the Russian Federation. The indicators of population and GDP are selected to reflect the overall economic development level of each federal subjects. The indicator of fixed capital investment is used to study the amount of economic activities such as the construction and fixed assets purchase of each federal subjects. The indicator of economic fixed assets is used to discuss the ability of enterprises to produce economic benefits from their production and operation activities in each federal subjects. The indicator of retail trade turnover is selected to study the level of goods and services sold by federal subjects through public trading platforms[1-2,10]. 

where KPi is the grade index of population. KEi is the grade index of GDP. KCi is the grade index of the fixed capital investment. KRi is the grade index of the economic fixed assets. KTi is the grade index of the retail trade turnover. KEi, KCi, KRi, KTi are calculated as KPi. Kti is the comprehensive economic grade index of the Russian federal subjects. Kei is the average economic grade index of the Russian federal subjects. n is the number of Russian federal subjects. According to the natural discontinuity classification method, the regional economic grades of the Russian federal subjects are divided into five classes: first class (Kei between 5.65~12.94), second class (Kei between 2.79~5.64), third class (Kei between 1.54~2.78), forth class (Kei between 0.65~1.53) and fifth class (Kei between 0.05~0.64).

Global trend analysis tool

Using the global trend analysis tool in ArcGIS, this paper studies the overall characteristics of the economic differences of each federal subjects in the whole of the Russian Federation [1-2,10]. Firstly, this paper draws the position of each federal subject on the X-dimensional plane and Y-dimensional plane. Then, it projects the per capita GDP value of each federal subject onto the X-Y orthogonal plane and Y-Z orthogonal plane respectively. Next, based on the scatter diagrams projected on the X-Y plane and Y-Z plane, this paper uses the second-order polynomial for spatial best fitting. Finally, from a macro perspective, this paper analyzes the overall change trend of East-West and North-South economic differences of the Russian Federation during 2002-2020. The X-axis represents the east-west direction of the whole territory of Russia (the arrow points to the East), and the Y-axis represents the north-south direction of the whole territory of Russia (the arrow points to the North). The height of each vertical line of the Z-axis represents the per capita GDP of each federal subject.

Spatial autocorrelation model

The spatial autocorrelation model is used to analyze the economic spatial agglomeration mode, economic correlation structure and economic differentiation characteristics of the adjacent subjects in the Russian Federation. Spatial autocorrelation refers to the correlation of the same kind of variables in different spatial positions. It is a measure of the aggregation degree of attribute values of spatial units. It could represent the spatial interaction, spatial diffusion and spatial dependence between variable data at a certain location and variable data at other locations. It contains the global spatial autocorrelation and the local spatial autocorrelation[1-2,21]. 

Global spatial autocorrelation is used to study the overall situation of spatial correlation and difference degree of unit attribute values in adjacent areas in the whole study area. In this paper, Moran index I is used to measure the degree of global spatial autocorrelation. 

Local spatial autocorrelation is used to study the differences of regional economic space in local scope, explaining whether there was spatial clustering and other correlation between the attribute values of local units and their adjacent units. In this paper, Getis-Ord Gi* is used to measure the degree of local spatial autocorrelation. It could describe the spatial difference pattern among cold spots and hot spots, exploring its pattern difference characteristics. 

Thanks!

Reviewer #2comments5: 

Comparison of own results with previous literature would also be needed when presenting the results.

Response: Thanks for your valuable comments and suggestions. Due to the update of data and the replacement of methods, we recalculate the Russian economic development differences. And we redraw the pictures about the temporal and spatial pattern of Russia's economic development. Based on the original manuscript, we raise many new lessons, consequences, and recommendations about the Russian economic development compared with the previous literatures in the results section. Please see the revised version of manuscript for details. 

Thanks!

Reviewer #2comments6: 

What would be the recommendation for policy makers in Russia? How to change their current policies in line of the results?

Response: Thanks for your valuable comments and suggestions. We suggest the recommendation for policy makers in Russia in the discussion section. The details are as follows:

The Russian Federation has vast territory and a large number of federal subjects. It has uneven population distribution and complicated ethnic issues. Under the background of two rounds of economic crisis in 2008 and 2014, the unbalanced regional economic development was not helpful to the political, economic, and social integration of the entire Russian Federation. At present, the Russian Federation has gradually got rid of the economic recession. However, its economic development still suffers the sanctions of European and American economies. It still has a long way to achieve balanced regional economic and social development.

 First, the Russian Federation needs reform the political systems and economic systems. It needs to establish long-term and short-term economic and social development strategies at different scales, such as the Russian Federation, the Federal Districts, the Federal Subjects, the towns and the rural areas. It needs to continue to deepen the innovation development strategy of the “Russian 2020 Development Strategy” and the economic modernization plan of the “Basic Principles of Anti-crisis Action in 2010”, relying on its domestic resources and intelligence advantages. Its economic development model should gradually transit from an energy resource export type to an innovative type. It needs to focus on promoting the development of the manufacturing industry, improving the business environment and enhancing the investment attraction. It should implement a new round of financial tax reform, optimizing the structure of state-owned assets. It should guarantee the SMEs to develop the real economy, limiting the inflation rate. It also should promote employment rate, increasing the actual income of residents. It needs to establish a unified social and economic space and it should implement a relatively equitable distribution system by effective production distribution and reasonable labor division.

 Second, the Russian Federation needs to implement unbalanced economic development strategies concerning to local natural conditions. The federal subjects in the western part of the Russian Federation continue to develop its traditional competitive advantages in energy resources. These federal subjects should build an energy price mechanism that is keeping with the international standards. And they should develop their potential advantages, increasing investment in high-tech fields. On the basis of realizing industrialization, they should upgrade their industrial structures. The federal subjects in the eastern part of the Russian Federation continue to focus on the economic development in resource-rich areas. These federal subjects should speed up the construction of energy transportation networking, improving the electricity infrastructure. They should apply high-new technology to create a high return rate on the economy, so as to absorb the return of labor and capital assets. They need to implement the diversified industrial development modes, improving the single structure of traditional energy resources.

 Third, the Russian Federation should actively participate in the cooperation of multilateral economies in Asia-Pacific. It should improve its opening-up policies, participating in China's "the Belt and Road", "China- Mongolia-Russia Economic Corridor," and "Changchun-Jilin-Tumenjiang Development and Opening Pilot Zone" and other initiatives. With the advantages of geographical proximity and resources complementary, it should build a free trade zone in the Tumen River Delta to increase its participating efficiency in the international labor division. It should promulgate some preferential policies to attract investment of China, Japan and Korea to participate in its energy bases development by building the oil and gas export channels in the Siberian and Far East Federal Districts. It should actively participate in some transport and energy projects such as China's high-speed rail items. And then it should broaden the diversified cooperation structures of modern agriculture, manufacturing, construction, transportation and tourism, etc.

 Forth, the Russian Federation should improve the transnational population migration policies. Its increasingly aging population and labor shortage have become the important constraints in the economic development in the Far East Federal District and other regions. It must implement ultra-conventional preferential policies to achieve the economic recovery and long-term stability in these regions. Every federal districts needs to stabilize the local population, preventing the population flow from the eastern part to the western part of the Russian Federation. The Russian Federation should promulgate more preferential policies to attract population migration to the Siberian and Far East Federal Districts. It should abandon the conservative xenophobia. It should promulgate more open and flexible immigration policies, creating a favorable investment environment to attract high-quality talents abroad. And it should create several economic growth poles in the Siberian and Far East Federal Districts.

Fifth, in recent years, with the economic rapid growth of the Asia-Pacific, Russian Federation has approved the "Society and Economic Development Plan for the Far East and Baikal Region" and the "Development Plan for the Border Areas of the Far East Federal District and Baikal Region" during 2014-2015. It has great strategic significance to develop the Far East Federal District and the Baikal region. First, developing the Far East and Baikal region can cope with the economic sanctions from the European and American countries, breaking through their export blockade. Second, developing the Far East Federal District and Baikal regions can guarantee the geopolitical security and prevent population loss of the Russian Federation. Third, developing the Far East Federal District and Baikal region can serve as a new economic growth point to solve the uneven economic development of the eastern and western areas in the Russian Federation. Forth, developing the Far East Federal District and Baikal regions meets China’s “Belt and Road Initiative”. The Far East Federal District and Baikal regions will become key areas for the opening of the Russian Federation to the Asia Pacific.

Thanks!

Reviewer #2comments7: 

What about future research ideas?

Response: Thanks for your valuable comments and suggestions. We propose the future research ideas in the discussion section. The details are as follows:

The Russian Federation is an important neighboring country of China. Under the background of "the Belt and Road" and "the economic corridor of China, Mongolia and Russia" initiatives, bilateral relations between China and Russia have reached a high level of cooperation. In 2020, the trade volume between China and Russia reached 107.8 billion dollars. China has become Russia's largest trading partner for many years. In the future, the economic linkage strength and its pattern between China and Russia will become important research ideas. From the perspective of people flow, based on the modified models of population geographical concentration and population quotient, we will study the overall situation of cross-border labor market, labor migration and mobility intensity, and their impact on the local employment market between China and Russia. And we will also study the quantity, structure and behavior characteristics of cross-border tourists, the source and destination of cross-border tourists, and the impact of tourism activities on local prices, consumption, housing, culture, etc., so as to analyze the characteristics and process of people flow. From the perspective of economic flow, combined with the revised models of urban flow, economic linkage strength and geo-economic relations, we will study the import and export commodity structure, trade flow and direction, trade structure differences at border ports between China and Russia. And we will also study the supply and demand potential, spatial distance and transportation cost, economic interaction and economic radiation intensity of cities, so as to explore the characteristics and process of economic flow. From the perspective of traffic flow, we will use the normalized modified accessibility coefficient to calculate the relative value and dynamic change of accessibility of cities between China and Russia. We will use the weighted travel time, economic potential and daily accessibility to calculate e the improvement degree of accessibility of cities between China and Russia. We will use the transportation connection strength model to study the strength of transportation connection function of cities between China and Russia, so as to study the characteristics and process of traffic flow. From the perspective of comprehensive flow, we will give weights to the matrices of people flow, economic flow and traffic flow. Through a series of algorithms to obtain the comprehensive flow matrix, we will calculate the spatial comprehensive linkage strength and evolution between China and Russia from the perspective of multi-dimensional factor flow, so as to summarize the characteristics and process of comprehensive flow.

Thanks once more!

---

## [Decision Letter · Decision Letter 1]

18 Jan 2022

Evolution characteristics of temporal and spatial pattern of Russian economic differences since the 21st century

PONE-D-21-11554R1

Dear Dr. Chu,

We’re pleased to inform you that your manuscript has been judged scientifically suitable for publication and will be formally accepted for publication once it meets all outstanding technical requirements.

Kind regards,

László Vasa, PhD

Academic Editor

PLOS ONE

Additional Editor Comments (optional):

Reviewers' comments:

Reviewer's Responses to Questions

**Comments to the Author**

1. If the authors have adequately addressed your comments raised in a previous round of review and you feel that this manuscript is now acceptable for publication, you may indicate that here to bypass the “Comments to the Author” section, enter your conflict of interest statement in the “Confidential to Editor” section, and submit your "Accept" recommendation.

Reviewer #1: All comments have been addressed

2. Is the manuscript technically sound, and do the data support the conclusions?

Reviewer #1: Yes

3. Has the statistical analysis been performed appropriately and rigorously? 

Reviewer #1: Yes

4. Have the authors made all data underlying the findings in their manuscript fully available?

Reviewer #1: Yes

5. Is the manuscript presented in an intelligible fashion and written in standard English?

Reviewer #1: Yes

6. Review Comments to the Author

Reviewer #1: The authors revised and improved their paper based on the reviewers' recommendations. In its current form, the paper is eligible for publication in PLoS ONE. The finally clarified methodology and context help us understand how the paper is contributing to the existing knowledge and in which terms it is original and worth for publishing.

7. PLOS authors have the option to publish the peer review history of their article (what does this mean?). If published, this will include your full peer review and any attached files.

Reviewer #1: No

---

## [Editor Report · Acceptance letter]

22 Mar 2022

PONE-D-21-11554R1 

Evolution characteristics of temporal and spatial pattern of Russian economic differences since the 21st century 

Dear Dr. Chu:

I'm pleased to inform you that your manuscript has been deemed suitable for publication in PLOS ONE. Congratulations! Your manuscript is now with our production department. 

Kind regards, 

on behalf of

Prof. Dr. László Vasa 

Academic Editor

PLOS ONE